# Integral approach to biomacromolecular structure by analytical-ultracentrifugation and small-angle scattering

Ken Morishima [1], Aya Okuda[1], Rintaro Inoue[1], Nobuhiro Sato[1], Yosuke Miyamoto[1], Reiko Urade[1], Maho Yagi-Utsumi [2,3,4], Koichi Kato [2,3,4], Rina Hirano[5,6], Tomoya Kujirai[5], Hitoshi Kurumizaka [5,6] & Masaaki Sugiyama [1✉]

Currently, a sample for small-angle scattering (SAS) is usually highly purified and looks monodispersed: The Guinier plot of its SAS intensity shows a fine straight line. However, it could include the slight aggregates which make the experimental SAS profile different from the monodispersed one. A concerted method with analytical-ultracentrifugation (AUC) and SAS, named as AUC-SAS, offers the precise scattering intensity of a concerned biomacro-molecule in solution even with aggregates as well that of a complex under an association-dissociation equilibrium. AUC-SAS overcomes an aggregation problem which has been an obstacle for SAS analysis and, furthermore, has a potential to lead to a structural analysis for a general multi-component system.

[1] Institute for Integrated Radiation and Nuclear Science, Kyoto University, 2-1010 Asashironishi, Kumatori, Sennan-gun, Osaka 590-0494, Japan. [2] Institute for Molecular Science (IMS), National Institutes of Natural Sciences, 5-1 Higashiyama, Myodaiji, Okazaki, Aichi 444-8787, Japan. [3] Exploratory Research Center on Life and Living Systems (ExCELLS), National Institutes of Natural Sciences, 5-1 Higashiyama, Myodaiji, Okazaki, Aichi 444-8787, Japan. [4] Graduate School of Pharmaceutical Sciences, Nagoya City University, 3-1 Tanabe-dori, Mizuho-ku, Nagoya, Aichi 467-8603, Japan. [5] Institute for Quantitative Biosciences, The University of Tokyo, 1-1-1, Yayoi, Bunkyo-ku, Tokyo 113-0032, Japan. [6] Department of Biological Sciences, Graduate School of Science, The University of Tokyo, 1-1-1 Yayoi, Bunkyo-ku, Tokyo 113-0032, Japan. ✉email: sugiyama@rri.kyoto-u.ac.jp

Structural investigations of biomacromolecules and their complexes in solution are essential to understanding physiological phenomena in biological systems. Several analytical methods have been developed and/or improved to address the investigations of such systems. Small-angle X-ray and neutron scatterings (SAXS or SANS) are classical techniques that give size information of particles in solution[1]. In addition, improved modern SAS gives further structural information: three-dimensional structure and structural fluctuation by combining scattering data with computational analyses, such as ab initio modeling[1,2] and molecular dynamics[3–5], respectively. These analyses require high-quality scattering data, i.e. the data purely from the target biomacromolecule.

There is an intrinsic obstacle to satisfy this requirement. Because SAS offers an ensemble-averaged scattering intensity of all particles in solution, unspecified aggregates in a solution (Supplementary Fig. 1a) will pollute the scattering intensity—especially the aggregates make abnormally upturn on the scattering intensity in the lower scattering angles (Supplementary Fig. 1b, c). In addition, a hidden problem has appeared[6,7]. We usually prepare for a purified sample in a structural analysis of a single molecule. However, even though the measured scattering intensity does not show such abnormal upturn and holds Guinier approximation[6], it could include a small amount of aggregates and they should be removed to obtain correct structure parameters[7]. In other words, it is difficult to judge that there still remains a small amount of aggregates in a solution only with SAS: A highly purified sample (Supplementary Fig. 2) looks nicely holds Guinier approximation but it included a small amount of aggregates. Practically, "revealing and removal of unspecified aggregates" has been one of the most significant challenges for SAS in many years.

The breakthrough for the "removal" is the development of size exclusion chromatography SAXS (SEC-SAXS)[7–9] which directly observes scattering of a size-separated particle solution eluting from a SEC-column. However, even with SEC-SAXS, we still have problems: demand of a relatively large amount of sample (>2 mg for a typical case), quick re-aggregation, and destruction of a weakly bound complex.

Analytical ultracentrifugation (AUC)[10] is an interesting technique for the "revealing" because it provides the concentration distribution of particles in solution with a small amount of sample (~0.05 mg) and is less destructive to complexes. Therefore, several previous studies used AUC to check the quality of SAS samples[11–13]. On the other hand, we have conceived in the advance to utilize AUC for the "removal": The SAS intensity of multi-component solution can be decomposed into those of components by utilizing the information of their concentration distribution obtained with AUC. In this way, the SAS intensity of a certain component, such as a monomer in solution, can be extracted from that of the whole solution even though the solution includes the unspecified aggregates.

By integral use of AUC and SAS, we have succeeded in extracting the SAS intensity of a specific monomer in a solution with also contained its aggregates. Furthermore, we have applied the method to obtain the SAS intensity of a weakly bound complex under an association–dissociation equilibrium. In this paper, we report on this newly developed "AUC-SAS" method.

## Results

**Extraction of monomer scattering (removing aggregation).** A scattering intensity $I_{mul}(q)$ ($q$: magnitude of scattering vector) of a multi-component system is represented by

$$I_{mul}(q) = ci_{mul}(q) = \sum_{j=1}^{n} c_j i_j(q), \qquad (1)$$

where $c$, $i_{mul}(q)$, and $n$ are total mass concentration, scattering intensity per $c$, and number of components, and $c_j$ and $i_j(q)$ are mass concentration and scattering intensity per $c_j$ for the $j$th component, respectively. Here, because $i_{mul}(q)$ is provided with SAS experiment as $i_{exp}(q)(= I_{exp}(q)/c)$ to solve a scattering intensity of a concerned component $k$, $i_k(q)$, the followings are required; "number of components, $n$", "concentrations of all components, $\{c_j\}$", and "scattering intensity(s) except for the concerned component, $i_j(q)$ $(j \neq k)$". AUC gives $n$ and $\{c_j\}$. Therefore, the remains are $i_j(q)$ $(j \neq k)$.

Firstly, our purpose is confined to extract the scattering intensity of the monomer ($k = 1$) from the scattering of the whole solution including unspecified aggregates ($j \geq 2$). Accordingly, the task is to figure out the scattering intensities of unspecified aggregates $i_j(q)$ $(j \geq 2)$. In general, it is difficult to know these scattering intensities, individually. Here, we notice that samples for a general SAS are highly purified, for which the following three conditions hold in most cases. Condition (i): Highly denatured contaminates have already been removed by purification. The remaining aggregates are simple oligomers with the low aggregation degree ($2 \leq j \leq 4$ at most) and their total weight fraction is less than ca 10%. Condition (ii): The inner structures of aggregates are identical with that of the monomer because the aggregates are assumed to be simple homo-oligomers, such as assembly of neat monomers. Condition (iii): Guinier approximation is established for experimental scattering intensity $ci_{exp}(q)$. This is the dangerous point because this could mislead us that the solution is monodispersed as described in "Introduction".

Under these conditions, we have developed a method, which extracts $i_1(q)$ from $i_{exp}(q)$ including the unspecified aggregates. Here, we explain the developed five-steps protocol shown in Fig. 1, taking a purified bovine serum albumin (BSA) solution as an example (see "Methods" and Supplementary Fig. 2).

Step 1: SAS measurement

Scattering intensity of multi-component system $ci_{mul}(q)$ is measured as $ci_{exp}(q)$ (open black circles in Fig. 2a). The mean gyration radius $R_{ge}$ and forward scattering intensity $ci_{exp}(0)$ are calculated with Guinier analysis (black circles and line in Fig. 2b and non-treated SAXS in Supplementary Table 1). If the scattering intensity deviates from the Guinier approximation (Supplementary Fig. 1b, c) or the obtained $R_{ge}$ is abnormally large, further purification should be applied.

Step 2: AUC measurement

Sedimentation velocity-AUC (SV-AUC)[14] is conducted for the same solution subjected to the SAS measurement. The measured sedimentation coefficient distribution $c(s_{20,w})$ offers $n$ and $\{c_j\}$ (Fig. 3, Supplementary Table 2).

Step 3: Forward scattering intensity of monomer, $i_1(0)$

The forward scattering intensity ratio $t_1$ of monomer $c_1 i_1(0)$ to the whole forward scattering $ci_{mul}(0)$ is calculated with AUC-measured $\{c_j\}$ as follows (Supplementary Note 1):

$$t_1 \equiv \frac{c_1 i_1(0)}{ci_{mul}(0)} = \frac{c_1}{\sum_{j=1}^{n} c_j j}. \qquad (2)$$

The $i_1(0)$ is calculated as $t_1 ci_{mul}(0)/c_1 = t_1 ci_{exp}(0)/c_1$: $ci_{mul}(0)$ is provided as $ci_{exp}(0)$ with a SAS experiment for a multi-component system as described before. Here, $ci_{exp}(0)$ and $c_1 i_1(0)$ are black and blue squares in Fig. 4, respectively, clearly indicating that a few % of aggregates (5.9% in the example, BSA1) generates the excess scattering $c_a i_a(0)$ $(= \sum_{j=2}^{4} c_j i_j(0)$ in the example) (red bar in Fig. 4) in the observed whole $ci_{exp}(0)$.

Step 4: Scattering of monomer in the high $q$, $i_{lh}(q_h)$

It is assumed that the aggregates are simple associated homo-oligomers (condition (ii)). In this case, an inner structure of the aggregates is same as a monomer. Therefore, the scattering

## AUC-SAS for multi-component system

$$ci_{\mathrm{mul}}(q) = \sum_{j=1}^{n} c_j i_j(q)$$

> Whole mass concentration, $c$
> Whole scattering intensity per mass concentration, $i_{\mathrm{mul}}(q)$
> Mass concentration of $j$-th component, $c_j$
> Scattering intensity per mass concentration of $j$-th component, $i_j(q)$

Concerned component, $k$ e.g. $k = 1$ : monomer in aggregation system
$k = \mathrm{AB}$ : AB complex in A + B ↔ AB system

**Step 1: SAS measurement**
Whole solution scattering intensity, $i_{\mathrm{exp}}(q)\ (= i_{\mathrm{mul}}(q))$
> + scattering intensity(s) of measurable component(s)
> $i_j(q)\ (j \neq k)$ (e.g. $i_A(q)$ and $i_B(q)$ )

→ - Mean gyration radius, $R_{\mathrm{ge}}$
 - Mean forward scattering, $i_{\mathrm{exp}}(0)$

**Step 2: AUC measurement**
Sedimentation coefficient distribution, $c(s_{20,\mathrm{w}})$ (SV-AUC)
or dissociation constant, $K_D$ (SE-AUC)

→ - Number of components, $n$
 - Mass Concentrations, $\{c_j\}$

$i_j(q)\ (j \neq k)$ obtained? **Yes**

**No** (e.g. aggregation system)

**Step 3: Calculation of forward scattering intensity, $i_1(0)$**
Forward scattering intensity ratio, $t_1 = c_1 i_1(0)/ci_{\mathrm{mul}}(0)$ : $t_1 = c_1/\sum_{j=1}^{n} c_j j$
→ $i_1(0) = t_1 c i_{\mathrm{exp}}(0)/c_1$

**Step 4: Estimation of scattering intensity in high $q_{\mathrm{h}}$, $i_{1\mathrm{h}}(q_{\mathrm{h}})^*$**
Inner structures of aggregates ($j \geq 2$ ) are identical to that of monomer ($k = 1$).
→ $i_{1\mathrm{h}}(q_{\mathrm{h}})^* \approx i_{\mathrm{exp}}(q_{\mathrm{h}})$

**Step 5A: Extraction of concerned scattering intensity, $i_1(q)$**

5A-1. Set of initial scattering intensity, $i_1(q)^*$
> In low $q_{\mathrm{l}}$: $i_{1\mathrm{l}}(q_{\mathrm{l}})^*$ $\left( = i_1(0)\exp\left[-(R_{\mathrm{g1}}^* q_{\mathrm{l}})^2/3\right] \right)$
> In high $q_{\mathrm{h}}$: $i_{1\mathrm{h}}(q_{\mathrm{h}})^*$

Smooth connection at $q_c$ ( $d\ln(i_{1\mathrm{l}}(q_c)*)/dq^2 = d\ln(i_{1\mathrm{h}}(q_c)*)/dq^2$ )
→ $i_1(q)^* = i_{1\mathrm{l}}\ (q \leq q_c)^* + i_{1\mathrm{h}}\ (q > q_c)^*$

5A-2. Refinement of scattering intensity, $\mathcal{R}\ (i_1(q)*)$

Refinement $\mathcal{R}$ with expanded Guinier formula
→ $i_1(q) = \mathcal{R}\ (i_1(q)*)$

**Step 5B: Extraction of concerned scattering intensity**
Calculation of scattering intensity with information of Steps1, 2
→ $i_k(q) = [ci_{\mathrm{exp}}(q) - \sum_{j \neq k} c_j i_j(q)]/c_k$

(e.g. association-dissociation equilibrium system)

**Fig. 1 Protocol of AUC-SAS for multi-component system.** For aggregation system, in Steps 3 and 4, the contribution of monomer in the scattering intensity is estimated because that of aggregation is not measurable directly. On the other hand, if all $i_j(q)\ (j \neq k)$ are obtained by individual measurements, such as an association–dissociation equilibrium system, Steps 3 and 4 are skipped.

intensity of monomer in the high $q$-range, $i_{1\mathrm{h}}(q_{\mathrm{h}})$, is almost identical to that of oligomer $i_{j\mathrm{h}}(q_{\mathrm{h}})(j \geq 2)$.

$$i_{1\mathrm{h}}(q_{\mathrm{h}}) \approx i_{\mathrm{mul}}(q_{\mathrm{h}}) = i_{\mathrm{exp}}(q_{\mathrm{h}}). \qquad (3)$$

It should be considered that the difference between $i_{1\mathrm{h}}(q_{\mathrm{h}})$ and $i_{\mathrm{mul}}(q_{\mathrm{h}})$ arises toward lower $q$-range. Here, the consideration is briefly described as follows (Supplementary Figs. 3 and 4, and Supplementary Note 2 in detail). To estimate the lowest $q^*$, where Eq. (3) holds, and to calculate the maximum difference at $q^*$, the intensity ratio $r(q)$ of whole scattering intensity to that of monomer is introduced as follows:

$$r(q) \equiv \frac{i_{\mathrm{mul}}(q)}{i_1(q)} = \frac{\left(\sum_{j=1}^{n} c_j i_j(q)\right)/c}{i_1(q)}. \qquad (4)$$

Here, two simple models for oligomers are introduced: linearly aligned and closed packing oligomerization models ($j \leq 4$; Supplementary Fig. 3). As the first assumption, the orientation of monomers in the oligomers is averaged and then their scattering intensities $i_j(q)(j \geq 2)$ (Eqs. (S7)–(S12) in Supplementary Note 2-1) are calculated based on Debye function with monomer scattering intensity $i_1(q)$ (Eqs. (S6) in Supplementary Note 2-1).

As shown in Supplementary Fig. 4, $r(q)$ with $i_j(q)$ and $\{c_j\}$ for the present example (BSA1) showed rapidly asymptotical approach to unity, and the deviation from unity is less than 1.8% in $q^* R_{\mathrm{g1}} \geq 1.0$, where $R_{\mathrm{g1}}$ is the gyration radius of the monomer. Therefore, in the case with a few % of aggregates (5.9% in the example, BSA1), it is approximated to be $i_{1\mathrm{h}}(q_{\mathrm{h}}) \approx i_{\mathrm{mul}}(q_{\mathrm{h}}) = i_{\mathrm{exp}}(q_{\mathrm{h}})$ within the maximum difference of 1.8% in

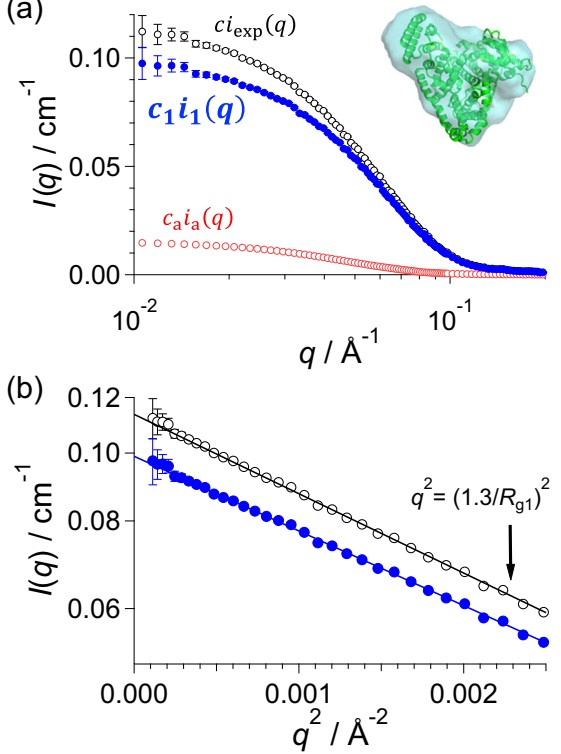

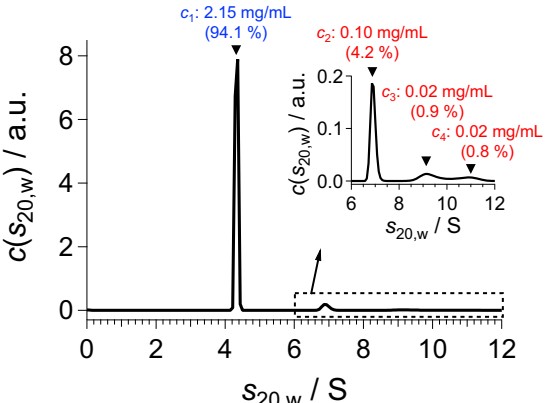

**Fig. 2 Extraction of SAXS intensity of monomer component for BSA1 by AUC-SAS. a** SAXS intensities and **b** Guinier plots for the BSA1 solution. Black, blue, and red circles indicate $ci_{exp}(q)$ (non-treated SAXS), $c_1i_1(q)$ and $c_ai_a(q)$ obtained with AUC-SAS as the final result, respectively. Solid lines represent the least-square fitting with Guinier formula. Inset shows an ab initio model based on $i_1(q)$.

**Fig. 3 Sedimentation coefficient distribution $c(s_{20,w})$ of the BSA1 solution obtained with SV-AUC.** There are four components: $c_1$, $c_2$, $c_3$, and $c_4$ represent the concentrations of the monomer, dimer, trimer, and tetramer, respectively. The weight fractions are also shown in the parentheses.

$q_h > q^* = 1.0/R_{g1}$ (Fig. 5a). Accordingly, in the present AUC-SAS protocol, the initial (tentative) scattering intensity of monomer in the high $q$-range $i_{1h}(q_h)^*$ is set to be $i_{exp}(q_h)$ ($q_h > q^*$): The closed blue circles in Fig. 5b represents $c_1i_{1h}(q_h)^*$ and open blue circles do extrapolation of $c_1i_{1h}(q)^*$ in the lower $q$-range ($q < q^*$).

Step 5A: Extraction of scattering intensity of monomer

To obtain the scattering intensity of monomer in whole $q$-range, it is necessary to find the scattering intensity in low $q$-range

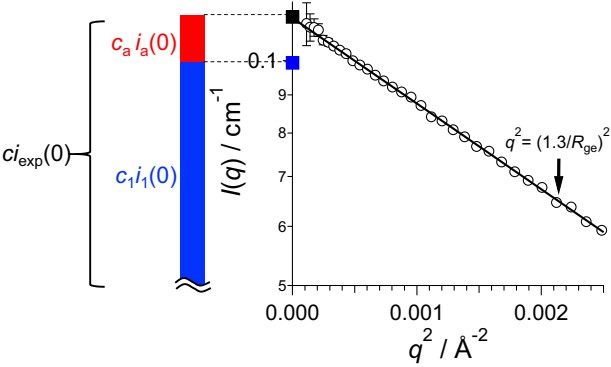

**Fig. 4 Guinier plot of $ci_{exp}(q)$ and the least-square fitting with Guinier approximation for the BSA1 measured by SAXS (open black circles and solid line, respectively).** Closed black and blue squares represent $ci_{exp}(0)$ and $c_1i_1(0)$, respectively. The bar graph expresses the contributions of monomer $c_1i_1(0)$ (blue part) and aggregates $c_ai_a(0)$ (red part) in $ci_{exp}(0)$.

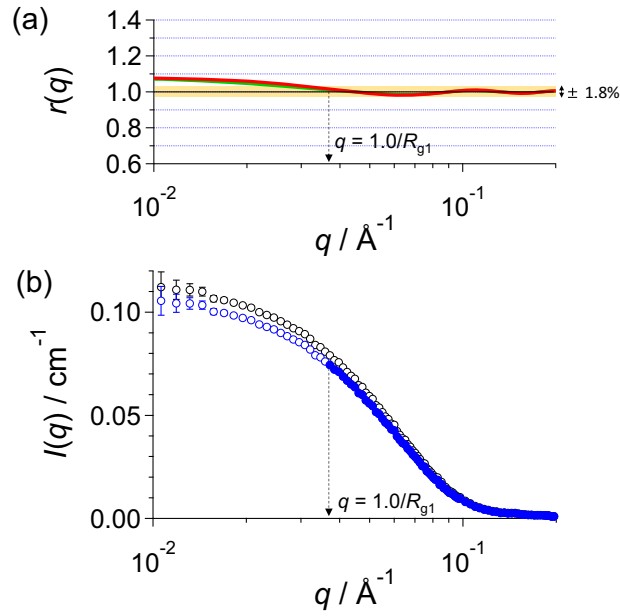

**Fig. 5 Estimation of $q$-range holding eq. (3) and its scattering intensity $c_1i_{1h}(q_h)^*$. a** Scattering intensity ratios $r(q)$ at $R_{g1} = 27.2$ Å with the $\{c_j\}$ for the present example (BSA1) (green: linearly aligned model and red: closed packing model: Supplementary Note 2). **b** Scattering intensities $ci_{exp}(q)$ (open black circles), $c_1i_{1h}(q_h)^*$ (closed blue circles), and extrapolation of $c_1i_{1h}(q)^*$ in the lower $q$-range (open blue circles).

$i_{1l}(q_l)$ filling the gap between $i_1(0)$ calculated in Step 3 and $i_{1h}(q_h)^* (= i_{exp}(q_h))$ set in Step 4.

5A-1. Setting the initial scattering intensity, $i_1(q)^*$:

In general, $i_{1l}(q)$ should satisfy Guinier approximation, holding the $y$-intercept of $i_1(0)$ ($= t_1ci_{mul}(0)/c_1$). Therefore, the initial scattering intensity in low $q$-range $i_{1l}(q_l)^*$ is semi-empirically set as follows:

$$i_{1l}(q_l)^* = i_1(0)\exp\left[-\frac{\left(R_{g1}^*q_l\right)^2}{3}\right]. \quad (5)$$

Here, the initial gyration radius of monomer $R_{g1}^*$ is chosen for making a smooth connection of $i_{1l}(q_l)^*$ to $i_{1h}(q_h)^*$ at the connection point $q_c$: $d\ln(i_{1l}(q_c)^*)/dq^2 = d\ln(i_{1h}(q_c)^*)/dq^2$

(Supplementary Note 3-1). Then, the initial whole scattering intensity of monomer $i_1(q)^*$ is provided with $i_{1l}(q_l)^*$ ($q_l < q_c$) and $i_{1h}(q_h)$ ($q_h \geq q_c$) (Supplementary Fig. 5).

5A-2. Refinement of the whole scattering intensity, $\mathcal{R}(i_1(q)^*)$:

There is a possibility that the initial whole scattering intensity $i_1(q)^*$ involves errors caused from the semi-empirically obtained $i_{1h}(q_h)^*$ and $i_{1l}(q_1)^*$. In order to refine $i_1(q)^*$, we utilized the expanded Guinier formula which holds to a relatively higher $q$-range: The expanded Guinier formula is prepared with the polynomial expansion of Debye formula[15] (Supplementary Figs. 6 and 7, Supplementary Notes 3-2 and 3-3). The final result $i_1(q)$, the full scattering profile of monomer extracted by AUC-SAS, is shown with closed blue circles in Fig. 2a and Supplementary Fig. 8. The structural parameters extracted by the above procedure completely agree with those from SEC-SAXS as listed in Supplementary Table 1, and the derived three-dimension ab initio model[2] well reproduced the crystal structure (Fig. 2a). Furthermore, AUC-SAS also succeeded to extract the scatting intensities of target protein in the other samples (ovalbumin and apoferritin: Supplementary Figs. 9 and 10, Supplementary Tables 1 and 2).

The AUC-SAS has advantages over the SEC-SAXS to obtain the monomer scattering intensity from solution including aggregates. A required sample amount for AUC-SAS is 0.1–0.25 mg of proteins (1–3 mg/mL in 30 μL for SAXS and 50 μL for SV-AUC) whereas that even for the recent high performance SEC-SAXS (https://www-ssrl.slac.stanford.edu/smb-saxs/content/documentation/sec-saxs/introduction) is 0.2–0.5 mg of proteins (4–10 mg/mL in 50 μL). Furthermore, AUC-SAS has better resolution for molecular separation than SEC and does not make it problem that the monomers quickly re-assemble after the separation with SEC. However, there is also limitation to provide correct result with the present AUC-SAS protocol about the upper concentration of aggregates around 12%. This is mentioned in Supplementary Note 4 (Supplementary Figs. 11–13 and Supplementary Table 3).

**Extraction of complex scattering.** AUC-SAS can extract the scattering intensity of a complex under an association–dissociation equilibrium. It should be noted that SEC-SAXS is unavailable to observe a weakly bound complex due to destruction of it by the SEC process (Supplementary Fig. 14). To the contrary, AUC has an ability to reveal concentration distribution of all components under an association–dissociation equilibrium even though the complex is weakly bounded.

Considering the equilibrium system of $A + B \leftrightarrow AB$, Eq. (1) is explicitly rewritten as

$$I_{exp}(q) = ci_{exp}(q) = c_A i_A(q) + c_B i_B(q) + c_{AB} i_{AB}(q). \quad (6)$$

AUC-SAS protocol is slightly modified to extract $i_{AB}(q)$ from $I_{exp}(q)$. Here, $i_A(q)$ and $i_B(q)$ are known with their individual SAXS measurements (Step 1) and then Steps 3 and 4 are skipped. Therefore, the key subject is to know the accurate concentrations for all components with AUC (Step 2) and combined analysis (Step 5B). Here, the modified protocol is demonstrated with an association–dissociation equilibrium system of hHR23b-UBL (component A) and PNGase-PUB (component B), which makes a weakly bounded complex[16].

Step 1: SAS measurement

$i_{exp}(q)$, $i_A(q)$, and $i_B(q)$ are individually measured with SAXS (Fig. 6a, Supplementary Fig. 15, and Supplementary Table 4).

Step 2: AUC measurement

SV-AUC cannot provide the concentrations of all components ($c_A$, $c_B$, and $c_{AB}$) for the system under the fast

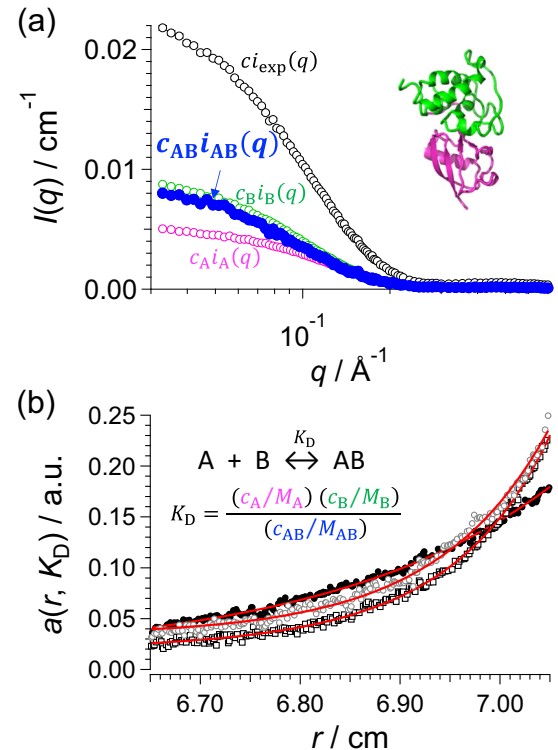

**Fig. 6 Extraction of SAXS intensity of complex by AUC-SAS. a** SAXS intensities for the hHR23b-UBL and PNGase-PUB system. Black, pink, green, and blue circles represent $ci_{exp}(q)$ of the experimental SAXS intensity for the mixture solution, $c_A i_A(q)$ for hHR23b-UBL, $c_B i_B(q)$ for PNGase-PUB by the individual experiments, and $c_{AB} i_{AB}(q)$ for the complex obtained with AUC-SAS as the final result, respectively. Inset shows the refined structure with normal mode analysis (https://files.inria.fr/NanoDFiles/Website/Software/Pepsi-SAXS/MacOS/Pepsi-SAXS-NMA)[22] (Supplementary Note 7). **b** Representative sedimentation equilibrium curves by SE-AUC at [hHR23b-UBL] = [PNGase-PUB] = 100 μM. Closed black circles, open black squares, and open gray circles represent the result at 20,000, 30,000, and 35,000 r.p.m., respectively. Red lines show the fitting curves (see "Methods").

association–dissociation process (Supplementary Fig. 16, Supplementary Note 5). Therefore, the concentrations of all components should be calculated with the dissociation constant $K_D$ by measured with the sedimentation equilibrium-AUC method (SE-AUC). Here, $K_D$ for the demonstration system were measured at three concentrations and with three rotation speeds by SE-AUC and $c_A$, $c_B$, and $c_{AB}$ were obtained (Fig. 6b, Supplementary Fig. 17, Supplementary Table 5, and Supplementary Note 6). Here, it is important that the absence of aggregates should be confirmed with SV-AUC (Supplementary Note 5) prior to SE-AUC measurements.

Step 5B: Extraction of scattering intensity of complex

The scattering intensity of complex $i_{AB}(q)$ (blue closed circles in Fig. 6a) is obtained from Eq. (6) with the intensities and concentrations obtained in Steps 1 and 2. The extracted scattering intensity was subjected to three-dimension modeling as well as the size analysis (Supplementary Fig. 18, Supplementary Table 4, Supplementary Note 7). This is the first report of the detailed structural information of this complex in solution.

## Discussion

Recently, SAS is required to study biomacromolecular structures in more complicated multi-component solution, for example, an association–dissociation equilibrium system involving aggregates.

As an advanced example for demonstrating this technique, the nucleosome was measured with AUC-SAS (Supplementary Note 8). The sample was highly purified but still included small amounts of aggregates, liberated histone complex, and DNA (Supplementary Fig. 19a, Supplementary Table 6). Therefore, both techniques, removing aggregation and complex scattering extraction, were required to find the precise scattering intensity of the nucleosome in solution. As shown in Supplementary Fig. 19b, c and Supplementary Table 7, AUC-SAS succeeded to provide the scattering intensity of nucleosome and, using it, the 3D-structure was reconstructed (Supplementary Fig. 20).

For many years, aggregation has been a real difficult obstacle in bio-SAS. AUC-SAS overcomes this aggregation problem and offers a precise scattering intensity of a concerned biomacro-molecule. In these days, the structural analysis of oligomers[17] in a multi-component solution attracts attention as well as that of monomer. For such cases, AUC-SAS is applicable if all scattering intensities except for the concerned component are individually available: the demonstration of AUC-SAS for nucleosome could be one of the examples (Supplementary Note 8). Furthermore, AUC-SAS also embraces structural analysis of a weakly bounded complex. In conclusion, AUC-SAS has a potential to become one of the standard methods to analyze structures of biomacromolecules in solution as shown in Supplementary Fig. 20.

Finally, we would like to remak the following. AUC-SAS does not require the very high-intensity beam for a sample-flow experiment such as SEC-SAXS. Therefore, AUC-SAS has a potential to be a complementary method for a laboratory-based SAXS and a standard SANS to synchrotron-based SEC-SAXS for structural analysis of biomacromolecules in solution.

## Methods

**Samples**. BSA (product# A2153), ovalbumin (OVA; product# A5503), and apo-ferritin (AF; product# A3641) purchased from Sigma Aldrich Co. were dissolved in 100 mM Tris/HCl (pH 7.5) buffer containing 100 mM NaCl. The solutions were purified by the anion-exchange chromatography with Resource Q 6 mL column (GE Healthcare) followed by the size exclusion chromatography with Superdex 200 increase 10/300GL column (GE Healthcare). The mass concentrations subjected to SAXS and AUC measurements were 2.29 mg/mL for BSA1 (used for demonstration of AUC-SAS protocol in the main text), 3.15 mg/mL for BSA2-4 (used for concentration boundary check in Supplementary Note 4), 2.00 mg/mL for OVA, and 1.81 mg/mL for AF, respectively. A quality of BSA1 was checked with SDS-PAGE. As shown in Supplementary Fig. 2a, no clear aggregation and contamination were observed in the sample.

The mixture solution of ubiquitin-like domain of the proteasome shuttle factor hHR23b (hHR23b-UBL; PDB code, 1P1A) and PUB domain of peptide:N-glycanase (PNGase-PUB; PDB code, 2D5U) were utilized as a demonstrated system which forms a weakly bounded complex under an association–disassociation equilibrium. The molecular weights of hHR23b-UBL and PNGase-PUB are 9.5 and 12.5 kDa, respectively. The expression and purification of the proteins have described previously[16]. The proteins were dissolved and dialyzed in 10 mM sodium phosphate buffer (pH7.0) (for all experiments) and PBS (only for SV-AUC). SE-AUC was carried out at three concentrations ([hHR23b-UBL] = [PNGase-PUB] = 100, 75, and 50 μM) and three rotation speeds (20,000, 30,000, and 35,000 r.p.m.). SAXS measurements are carried out at [hHR23b-UBL] = 200 μM, [PNGase-PUB] = 200 μM, and [hHR23b-UBL] = [PNGase-PUB] = 100 μM for mixture, where the intermolecular interference effect can be negligible on the scattering profile.

The nucleosome was prepared as the same manner in the previous report[18]. The sample was dialyzed against 20 mM Tris/HCl (pH7.5) buffer containing 50 mM NaCl and 1 mM DTT. The mass concentrations subjected to SAXS and AUC measurements were 1.29 mg/mL.

**Small-angle X-ray scattering**. All SAXS measurements were carried out with a laboratory-based instrument NANOPIX (Rigaku) equipped with high-brilliance point-focused generator of a Cu-Kα source (MicroMAX-007 HFMR, wavelength $(\lambda) = 1.54$ Å). The scattered X-rays were detected using a two-dimensional semi-conductor detector (HyPix-6000) with the spatial resolution of 100 μm. The sample-to-detector-distances were set to be 1280 mm (covered $q$-range: 0.010–0.20 Å$^{-1}$) for BSA, OVA, AF, and nucleosome experiments, and 355 mm (covered $q$-range: 0.030–0.70 Å$^{-1}$) for PNGase-PUB + hHR23b-UBL and nucleosome experiments, respectively. Two-dimensional scattering pattern was converted to a one-dimensional scattering intensity with SAngler software[19]. After the correction by the

transmittance and subtraction by the buffer scattering, the absolute scaled scattering intensity was obtained by referring to a standard scattering intensity of water $(1.632 \times 10^{-2}$ cm$^{-1})$[20]. All measurements were conducted at 25 °C.

**Analytical ultracentrifugation**. AUC measurements were conducted with a ProteomeLab XL-I analytical-ultracentrifuge (Beckman Coulter). The cell with a small volume (optical path: 1.5 mm, volume: 50 μL, Nanolytics) was used for the measurements. Two measuring methods, sedimentation velocity-AUC (SV-AUC) and sedimentation equilibrium-AUC (SE-AUC), were conducted depending upon the sample situations. The former measures sedimentation speed of particles and gives sedimentation coefficient distribution $c(s_{20,w})$. The later observes a concentration gradient under sedimentation equilibrium, which provides a dissociation constant $K_D$ for an association–dissociation system. The sample solutions were loaded at 50 μL for SV-AUC and 20 μL for SE-AUC. SV-AUC was performed using Rayleigh interference optics at 40,000 r.p.m. of rotor speed for BSA, OVA, AF, and nucleosome, and at 60,000 r.p.m. of rotor speed for hHR23b-UBL, PNGase-PUB, and their mixture, respectively. SE-AUC was carried out using absorbance optics at 20,000, 30,000, and 35,000 r.p.m. for PNGase-PUB and hHR23b-UBL. All measurements were conducted at 25 °C. The SV-AUC data were analyzed with SEDFIT (http://www.analyticalultracentrifugation.com/sedfit.htm) software which executed the fitting with Lamm formula[14]. The sedimentation coefficient was converted to the value at 20 °C in pure water $(s_{20,w})$. The molecular weight for each component was calculated using the peak $s_{20,w}$ and friction ratio $f_r$. The weight fraction $c_j$ for each component was obtained from the peak area. The SE-AUC data were analyzed with SEDPHAT (http://www.analyticalultracentrifugation.com/sedphat/default. htm) software, which conducts the fitting to the SE-AUC results with the association–dissociation equilibrium model as follows[21]:

$$a(r, K_D) = \varepsilon_A c_A(r_0)\exp\left[\frac{\omega^2 M_A}{2RT}(1 - \bar{v}_A\rho)(r^2 - r_0^2)\right]$$
$$+ \varepsilon_B c_B(r_0)\exp\left[\frac{\omega^2 M_B}{2RT}(1 - \bar{v}_B\rho)(r^2 - r_0^2)\right] \quad (7)$$
$$+ \varepsilon_{AB} c_A(r_0)c_B(r_0)\frac{M_{AB}}{M_A M_B K_D}\exp\left[\frac{\omega^2 M_{AB}}{2RT}(1 - \bar{v}_{AB}\rho)(r^2 - r_0^2)\right],$$

where $a(r, K_D)$ is an absorbance at radius $r$ for the equilibrium system with the dissociation constant $K_D$, $\varepsilon$ is the extinction coefficient, $c(r_0)$ is the concentration at the reference radius $r_0$, $\omega$ is the angular velocity, $M$ is the molecular weight, $\bar{v}$ is the partial specific volumes, $\rho$ is the solvent density, and $RT$ is the multiplication of the gas constant and absolute temperature. For accurate determination of the free parameters, we carried out the global fitting analysis for the three different concentrations and three different rotation speeds with same $K_D$. For the analysis, the partial specific volumes $(\bar{v})$ of each protein were calculated from their amino acid sequences with SEDNTERP (http://www.jphilo.mailway.com/download.htm) software. The density and viscosity of solvents were measured with the density meter DMA4500M (Anton Paar) and the viscometer Lovis 2000 M/ME (Anton Paar), respectively.

**Statistics and reproducibility**. The fittings for "Guinier formula" to derive $i(0)$ and $R_g$ were performed with the linear least-square method by Igor Pro (7.04) and the fitting for SE-AUC to derive $K_D$ was done with the non-linear least-square method (Levenberg–Marquardt algorithm) also by Igor Pro (7.04). The errors were calculated considering the error propagation theory with the $\chi^2$, which are listed in Supplementary Tables 1, 3, 4 and 7.

**Reporting summary**. Further information on research design is available in the Nature Research Reporting Summary linked to this article.

## Data availability

Source data for Figs. 2, 3, 6a, b are included in Supplementary Data 1–4, respectively. Any other data in the supplementary materials are available from the authors upon reasonable request.

## Code availability

All software used in this study is publicly available at the URLs below.
Igor Pro 7.04: https://www.wavemetrics.net/previous7.html
SAngler 2.0.0: http://pfwww.kek.jp/saxs/SAngler.html
SEDFIT 15.01c: http://www.analyticalultracentrifugation.com/sedfit.htm.
SEDPHAT 14.0: http://www.analyticalultracentrifugation.com/sedphat/default.htm.
SEDNTERP 1.10: http://www.jphilo.mailway.com/download.htm.
Pepsi-SAXS-NMA: https://files.inria.fr/NanoDFiles/Website/Software/Pepsi-SAXS/MacOS/Pepsi-SAXS-NMA
GNOM: https://www.embl-hamburg.de/biosaxs/gnom.html
DAMMIN: https://www.embl-hamburg.de/biosaxs/dammin.html

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

## Acknowledgements

We especially wish to thank to Prof. Susumu Uchiyama (Osaka University) for suggestive and fruitful comments on the analysis of AUC and also would like to thank to Dr. Soji Murayama (Beckman Coulter) for kind help for instruction of AUC. We would like to expree our appliciation to Prof. Martin E. Vigild (Thechical University of Denmark) for his comments on SAXS data analysis. We also thank Ms. Yukiko Isono (IMS/ExCELLS) for protein purification. Preliminary SAXS experiments at Photon Factory were performed under Proposal No. 2017G100 and 2018G680. This work was supported by MEXT/JSPS KAKENHI Grant Numbers (JP19K16088 to K.M., JP17K07361, JP19KK0071, and JP20K06579 to R.I., JP17K07816 to N.S., JP18H05229, JP18H05534, and JP18H03681 to M.S.) and by the Future Development Funding Program ofKyoto University Research Coordination Alliance, Research Fund for Young Scientists in Kyoto University and Fund for Project Research in Institute for Integrated Radiation and Nuclear Science, Kyoto University (KURNS). This work was also partially supported by the project for Construction of the basis for the advanced materials science and analytical study by the innovative use of quantum beam and nuclear sciences in KURNS and by Joint Research by IMS (IMS program No. 205) and by the Platform Project for Supporting Drug Discovery and Life Science Research (Basis for Supporting Innovative Drug Discovery and Life Science Research (BINDS)) from AMED under Grant Number (JP20am0101076 to H.K.).

## Author contributions

K.M., A.O., Y.M., R.I., N.S., R.U., and M.S. performed SAXS measurements and analysis. K.M., Y.M., and M.S. performed AUC measurements and analysis. M.Y.-U. and K.K. prepared hHR23b-UBL and PNGase-PUB. R.H, T.K., and H.K. prepared H2A nucleosome. K.M. and M.S. designed research and all the authors wrote the paper.

## Competing interests

The authors declare no competing interests.
