## [Peer Review File · Communications Biology]

Reviewers' comments:

Reviewer #1 (Remarks to the Author):

The authors have presented a nice, clear and easily applied method for using analytical ultracentrifugation data to deconvolute aggregated/polydisperse SAXS data. The data they provide indicates that the technique is largely supported, and that it can be used to recover scattering curves from such systems. I have no concerns with the SAXS data itself, or the analysis thereof.

I do have a couple of comments in regards to some of the rhetoric used to justify why such a method is required. The authors state that large amounts of protein are required for SEC-SAXS (they state > 2mg). This is not strictly true. At our beamline we typically run 0.2 - 0.5 mg in 50 microliters on SEC-SAXS measurements, even for small and aggregation prone samples. And while I agree that SEC can disrupt weakly bound complexes, AUC will also do the same thing, particularly in sedimentation velocity measurements. Most often this results in a broad sedimentation coefficient distribution, and inaccurate proportions of the various populations. In Equilibrium measurements, the sample doesn't reliably reach equilibrium, as the complex tends to drift between associated and dissociated states.

Further 50 microliters of sample volume in a standard AUC centrifuge cell is pushing the limits for a useful range of radii in a velocity experiment - it is much more typical to use 300 microliters.

I am also concerned about the reproducibility of the various populations from preparation to preparation. The authors indicate that the same preparation/batch was used for SAXS as for AUC, but for access to SAXS, particularly at synchrotron facilities, timing can be an issue. So can I do the AUC experiment now and then collect SAXS data on a completely separate batch/preparation of material 6 months later and still apply this approach? If so then this is much more applicable.

I have no minor concerns - the paper was well written and clear.

Reviewer #2 (Remarks to the Author):

In this manuscript, the authors use a new combined experimental technique in the field to obtain precise scattering intensity of biomacromolecules in solution. A combined AUC-SAS analysis is claimed to be a better and more accurate substitute to the current state of the art of utilizing a combination of SEC-SAXS.

I do believe this is a promising technique and will be an important addition in this research field. I do recommend the manuscript for publication after making some changes to make it a better read.

However, the concept of combining AUC with other techniques is an ongoing research area and I would advise the authors to cite a few recent publications where AUC is combined with other techniques for orthogonal characterization, especially for nanoparticles.

I had to read quite a few times to understand the algorithm. The section on extracting monomer scattering needs a lot of refinement. It is difficult to shift between the main manuscript and the supplementary information frequently. There is a lot of important information in the SI. Unless there is a restriction of length for the main manuscript, important details like the use of forward scattering, approximations in the high q range etc. should be part of the main manuscript. Please make sure all the variables used are properly explained. The manuscript requires thorough check for typographical errors and good editing.

Two main assumptions are low aggregation degree and the total weight fraction of aggregates to

be on the lower side (given approximately as 10 %). The authors need to justify this. The examples given have aggregation degree and relatively weight percentage of aggregates much lower than these values. They need to elaborate more on these concepts and possibly give a range in which the proposed method would work.

Reviewer #3 (Remarks to the Author):

The paper 'Integral approach to biomacromolecular structure by analytical-ultracentrifugation and small-angle scattering' by K. Morishima et al. presents a semi-empirical approach to correct SAS curves from biological molecules in solutions regarding the aggregation effect. The correction is made basing on an additional measurement of partial fractions of aggregates by analytical ultracentrifuging. This way is considered to some extent as an alternative to SEC-SAXS. It is demonstrated that the approach works well for several protein solutions, which is proved by consistency with the data of SEC-SAXS. The motivation is claimed well. The problem of the protein aggregation in SAS experiments is old. The problem is actual and not only from the point of view of parasitic effects. See, for example, a very recent publication on the study of oligomeric solutions of proteins at the pre-crystallization stage by SAXS [Marchenkova MA, et al., J Biomol Struct Dyn. 2019. doi: 10.1080/07391102.2019.1649195].

General remark

I am afraid the authors have not fully convinced me that the approach proposed is general and works well in all cases, not only for the solutions considered in the paper.

The approach lacks substantial mathematical background which should first be presented and discussed in details, may be, in a more specialized journal such as Journal of Applied Crystallography before it is considered to be a generalized method (such as, for example, indirect Fourier transform). I would definitely support the approach if it were used as a complementary part of some complex characterization of protein solutions close to those of BSA. Here, I have an impression that numerical solutions of partial problems are extended to a general case with avoiding a consideration of the limiting or boundary cases when the application of this approach starts to be problematic. The partial scattered intensities from aggregate populations are not an orthogonal system, which means that the solutions of the problems related to the series expansion depend on criteria used to judge about correctness and provide stability of the solutions regarding experimental errors. I did not find such kind of criteria in the work. There is no such problem with SEC-SAXS where the partial contributions are directly measured.

So, my recommendation is somewhat between 'decline' and 'major revision'.

Concise remarks

1) From the beginning, the authors use an extrapolation procedure for low q -values which is not discussed in the paper. However, the result depends on the smoothness of the scattered intensity in 'high q -value interval'. This smoothness is implicitly assumed for experimental data. Also, it depends on the choice of the minimum cut-off q -value of the 'high q -value interval'. In other words, the procedure is model dependent and, more important, affected by the choice of the q -value where the 'high q -value interval' starts. In Supplementary one can find the calculation of the simplest structure-factor for spherical dimers for one (!) size-value to judge about the aggregation effect on the choice of this interval. No clear criteria based on mathematical consideration are given. It is important that for the general case, such consideration should take into account the restrictions on the finite q -range, as well as possible effects of statistics and background (in case when the absolute scattering is not so high).

2) I am not sure that the proposed procedure for 'connecting' Guinier region with 'high q -value

interval' works well for the molecules with an inhomogeneous inner structure under low contrast. In the general case, there can be a parabolic dependence of $I(q)$ at q approaching zero with a positive coefficient (see, for example, the asymptotic behavior of the basic scattering functions in contrast variation approach in M.V.Avdeev, J. Appl. Cryst. 40, 2007, 56). So, the limits on the homogeneous approximation for single units in the aggregates should be discussed and specified, if possible.

3) I appreciate the comparison with SEC-SAXS. It is a strong evidence for the correctness of the proposed approach. Still, it would be more convincing from the point of view of generalization if the authors make separate simulations of experimental data for model aggregated units of spherical shape with different error levels in typical experimental q -ranges for different unit size scales and treat them in the way proposed. From the comparison with the exact scattering known, one can analyze the systematic deviations of the solutions and their stability with respect to the experimental errors.

Reviewer #4 (Remarks to the Author):

This manuscript by Morishima et al. presented a nice method for SAS data analysis by taking advantage of the information of the species in solution from analytical ultracentrifugation (AUC), the AUC-SAS method. Because the experimental conditions of both methods can be the same, the detailed information about the solution composition from AUC, particularly the size and fraction of the aggregates can seemly offer reliable constraints to the data analysis of SAS. This method can be applied to both non-interacting and interacting systems.

For non-interacting system, the key points of this approach are (1) to study the heterogeneous samples with sedimentation velocity analytical ultracentrifugation (SV-AUC); (2) to resolve the size distribution of the sample using the standard $c(s)$ analysis; (3). To decompose the SAS signal by utilizing the size/population information from SV-AUC. The authors also demonstrated that this approach can be applied to study interacting system, by characterization with sedimentation equilibrium in order to extract the fraction of the population for the unbound and bound species.

Overall, the manuscript is a clear and concise summary of the new AUC-SAS method.

Nevertheless, several points should be addressed to improve the clarity of the manuscript.

(1). Page4, "Step1..if the scattering intensity deviates from the Guinier approximation..".

What is the threshold/tolerance of the deviation?

(2). Page5: "Step2: AUC measurement"

The authors should describe why the two particular concentrations, 50 and 100 μM were used.

(3). How to distinguish the denatured contaminants and the oligomers from SAS and AUC data?

(4). There is some limitation of the application to study the weakly bounded complex in the current work.

(i). Sedimentation equilibrium data analysis is an ill-conditioned problem, which is very sensitive to degraded products or any contamination with small particles. Therefore, it is crucial to perform the experiment with multiple concentrations and multiple rotor speeds, so the concentration gradients can be globally analyzed to determine the population of the species with different sizes. In this study, only one rotor speed, 30,000 rpm was used. This limits the accuracy of the data analysis. At least, the authors should perform a reliable error analysis to determine the confidence intervals for K_d of the hetero-association.

(ii). Because of the intrinsic limitation of sedimentation equilibrium described above, the authors should consider further analysis for the SAS data using the min and max of the K_d , as well as floating it, to examine whether a better fit can be reached.

(5). In the main text, the authors should add a sentence to explain that this is a highly pure interacting system and aggregates are not observed from the individual proteins and their mixture. Therefore, the modeling can be simply based on mass action law.

Response to Reviewer #1:

The authors have presented a nice, clear and easily applied method for using analytical ultracentrifugation data to deconvolute aggregated/polydisperse SAXS data. The data they provide indicates that the technique is largely supported, and that it can be used to recover scattering curves from such systems. I have no concerns with the SAXS data itself, or the analysis thereof.

→ First of all, we sincerely appreciate the encouraging comment. The following comments have helped us to improve the paper.

I do have a couple of comments in regards to some of the rhetoric used to justify why such a method is required. The authors state that large amounts of protein are required for SEC-SAXS (they state $> 2\text{mg}$). This is not strictly true. At our beamline we typically run 0.2 - 0.5 mg in 50 microliters on SEC-SAXS measurements, even for small and aggregation prone samples.

→ Thank you for pointing it out. The relevant part (in page 3) is corrected as follows, " $> 2\text{mg}$ for typical case".

Furthermore, we added the following sentences in page 7.

"A required sample amount for AUC-SAS is 0.1 - 0.25 mg of proteins (1 - 3 mg/mL in 30 μL for SAXS and 50 μL for SV-AUC) whereas that even for the recent high performance SEC-SAXS¹⁵ is 0.2 - 0.5 mg of proteins (4 - 10 mg/mL in 50 μL)."

And while I agree that SEC can disrupt weakly bound complexes, AUC will also do the same thing, particularly in sedimentation velocity measurements. Most often this results in a broad sedimentation coefficient distribution, and inaccurate proportions of the various populations. In Equilibrium measurements, the sample doesn't reliably reach equilibrium, as the complex tends to drift between associated and dissociated states.

→ Thank you for the comment. In general, sedimentation velocity (SV)-AUC for fast association-dissociation equilibrium systems may not provide the concentrations of the components as you mentioned. Acutely, the demonstrated case in our study did not

provide the concentrations with SV-AUC. Then, we intended to obtain the concentrations from K_D measured with sedimentation equilibrium (SE)-AUC.

We also understood your concern about SE-AUC. It has been reported that the multi-measurement, a couple of concentration and rotation speeds, gives reliable K_D . (Vistica et al. Analytical biochemistry 326 (2004) 234-256). Therefore, in this revised version, we carried out the additional SE-AUC measurements with three concentrations and three rotation speeds to improve the reliability of K_D . In addition, to make the reliability of K_D clear, the fitting error of K_D was imposed in the error of extracted scattering intensity $c_{AB}i_{AB}(q)$ with considering the error propagation (Fig.6(a) and Table S4).

Further 50 microliters of sample volume in a standard AUC centrifuge cell is pushing the limits for a useful range of radii in a velocity experiment - it is much more typical to use 300 microliters.

→ Thank you for the helpful comment. To confirm the data reliability even with a small cell, we compared the sedimentation coefficient distribution $c(s_{20,w})$ measured with 50 μ L cell to that with 400 μ L cell. As shown in Fig.R1-1, both results are almost identical, indicating that the 50 μ L cell is well available at practical concentrations of

AUC-SAS (1 - 3 mg/mL). In addition, various small volume cells are recently developed with 3D printing and also provide reliable results (S.C. To, et al., Anal. Chem. 91 (2019) 5866-5873).

Fig.R1-1: The sedimentation coefficient distribution for the BSA solution at 1.0 mg/mL measured with 50 μ L (blue) and 400 μ L (red) cells.

I am also concerned about the reproducibility of the various populations from preparation to preparation. The authors indicate that the same preparation/batch was used for SAXS as for AUC, but for access to SAXS, particularly at synchrotron facilities, timing can be an issue. So can I do the AUC experiment now and then collect SAXS data on a completely separate batch/preparation of material 6 months later and still apply this approach? If so then this is much more applicable.

→ Thank you for the comment. We strongly recommend to measure **same batch** samples for AUC and SAXS *simultaneously*. However, we understand the situation; AUC is not always available beside a SAXS machine. If the simultaneous experiment is difficult, as an alternative, we recommend that the AUC measurements are performed before and after the SAXS measurement. AUC-SAS should be conducted only in the case that the AUC spectra are not changed.

I have no minor concerns - the paper was well written and clear.

→ Again, we appreciate your careful review and helpful comments.

Response to Reviewer #2:

In this manuscript, the authors use a new combined experimental technique in the field to obtain precise scattering intensity of biomacromolecules in solution. A combined AUC-SAS analysis is claimed to be a better and more accurate substitute to the current state of the art of utilizing a combination of SEC-SAXS.

I do believe this is a promising technique and will be an important addition in this research field. I do recommend the manuscript for publication after making some changes to make it a better read.

→ We would like to express our sincere appreciation to the reviewer for the encouraging comment. The following comments have helped us to improve the paper.

However, the concept of combining AUC with other techniques is an ongoing research area and I would advise the authors to cite a few recent publications where AUC is combined with other techniques for orthogonal characterization, especially for nanoparticles.

→ Thank you for the helpful advice. Following your comments, we cited the previous studies (ref. 12-14) which applied AUC and SAS to nanoparticles and biomacromolecules. In addition, the previous studies basically used AUC and SAS for the cross check of quality of samples in our knowledge. On the other hand, in our study, AUC and SAS were complementally used to extract the scattering intensity of the target component from that of multi-components system. Therefore, we added the following sentences with above mentioned citations (ref. 12-14) in page 2.

"Therefore, several previous studies used AUC to check the quality of SAS samples¹²⁻¹⁴. On the other hand, we have conceived in the advance to utilize AUC for the "removal": The SAS intensity of multi-component solution can be decomposed into those of components by utilizing the information of their concentration distribution obtained with AUC. "

I had to read quite a few times to understand the algorithm. The section on extracting monomer scattering needs a lot of refinement. It is difficult to shift between the main manuscript and the supplementary information frequently. There is a lot of important information in the SI. Unless there is a restriction of length for the main manuscript, important details like the use of forward scattering, approximations in the high q range etc. should be part of the main manuscript. Please make sure all the variables used are properly explained. The manuscript requires thorough check for typographical errors and good editing.

→ We are very sorry for difficulty to read the manuscript. We revised the explanation about the algorithm and re-arranged its order as follows.

1. Concerning about explanation of intensity in the high q range (new Step 4, please see below), we revised the paragraph of Step 4 to explain the mathematical background more. Even though the further detailed explanation was written in §S4 of SI, we believed that the main logic can be caught by reading the paragraph of Step 4 in the main manuscript.
2. New Figs.4 and 5 were moved from SI to the main manuscript.
3. To make the algorithm clearer, old Step 4 "Scattering profile in low q range" was divided into "Calculation of forward scattering intensity" and "Set of initial scattering intensity".
4. "Calculation of forward scattering intensity" was put on new Step 3, and "Set of initial scattering intensity" was also put on new Step 5A-1.
5. Then old Step 3 "Estimation of scattering intensity in high q range" was moved to new Step 4.
6. Figure 1 was redrawn to follow the above mentioned rearrangements of the algorithm.
7. Old Fig.2 were divided into two figures. New Figs. 2 and 3 were concerned about "Extraction of monomer scattering" and new Fig.6 was about "Extraction of scattering of weakly bounded complex".

This modification on the algorithm makes no effect on the final result.

Two main assumptions are low aggregation degree and the total weight fraction of aggregates to be on the lower side (given approximately as 10 %). The authors need to justify this. The examples given have aggregation degree and relatively weight percentage of aggregates much lower than these values. They need to elaborate more on these concepts and possibly give a range in which the proposed method would work.

→ Thank you for the helpful comments. Following the suggestion, the new section "Limitation and improvement of AUC-SAS" in SI-§S7 was added to discuss the applicable limitation in the aggregation degree (up to 4) and their total weight fractions. In addition, we presented the further possibilities of AUC-SAS to apply to a system with the higher concentration of aggregates.

In the "Limitation" part (§S7-1), AUC-SAS was tested for BSA solutions with various amounts of aggregates (BSA2-4). The present protocol offered the correct scattering intensity for the solution containing with 12.4 % of weight fraction of aggregates, while could not for the solutions with 18.6 % of weight fraction of aggregates (Fig.S11). Therefore, the present protocol is applicable up to 12 % and the boundary could locate ca. 15%.

In the "Improvement" part (§S7-2), we considered the reason to make the present protocol failed under the higher concentration of aggregates and then presented the possible improvement. The failure of the present protocol under the higher concentration of aggregates could arise from the inaccuracy of $i_{\text{exp}}(0)$ because the experimental scattering intensity does not hold Guinier approximation (see Fig.S11(d)). The inaccuracy of $i_{\text{exp}}(0)$ causes the wrong $i_1(0)$ which is an important value in Step 5A-1 of the preset AUC-protocol. As an improved approach to obtain more precise $i_1(0)$, we estimated $i_1(0)$ with eq.(S21). The improved approach led the consistent result with SEC-SAXS for the sample even with 18.6 % of weight fraction of aggregates as shown in Fig.S12. We will make the further discussion including other limitations and improvements in a next paper, for example, the breakage of the approximation of $i_{1h}(q) \approx i_{\text{exp}}(q)$ described in SI-§S4.

Response to Reviewer #3:

The paper 'Integral approach to biomacromolecular structure by analytical-ultracentrifugation and small-angle scattering' by K. Morishima et al. presents a semi-empirical approach to correct SAS curves from biological molecules in solutions regarding the aggregation effect. The correction is made basing on an additional measurement of partial fractions of aggregates by analytical ultra-centrifuging. This way is considered to some extent as an alternative to SEC-SAXS. It is demonstrated that the approach works well for several protein solutions, which is proved by consistency with the data of SEC-SAXS. The motivation is claimed well. The problem of the protein aggregation in SAS experiments is old. The problem is actual and not only from the point of view of parasitic effects. See, for example, a very recent publication on the study of oligomeric solutions of proteins at the pre-crystallization stage by SAXS [Marchenkova MA, et al., J Biomol Struct Dyn. 2019. doi: 10.1080/07391102.2019.1649195].

→ First of all, we sincerely appreciate your careful review and comments. All comments are helpful for revise of the manuscript.

After carefully reading the suggested paper, we recognize that there are meaningful aggregates in a multi-component solution in these days. In this study, we did not examine an AUC-SAS protocol with an oligomeric solution of proteins at the pre-crystallization stage in the suggested paper directly, but demonstrated the protocol for the similar system (nucleosome in SI-§S13). Therefore, we added following sentences with the reference you suggested in page 8.

"In these days, the structural analysis of oligomers¹⁷ in a multi-component solution attracts attention as well as that of monomer. For such cases, AUC-SAS is applicable if the all scattering intensities except for the concerned component are individually available: the demonstration of AUC-SAS for nucleosome could be one of examples (SI-§S13)."

General remark

I am afraid the authors have not fully convinced me that the approach proposed is general and works well in all cases, not only for the solutions considered in the paper.

The approach lacks substantial mathematical background which should first be presented and discussed in details, may be, in a more specialized journal such as Journal of Applied Crystallography before it is considered to be a generalized method (such as, for example, indirect Fourier transform). I would definitely support the approach if it were used as a complementary part of some complex characterization of protein solutions close to those of BSA. Here, I have an impression that numerical solutions of partial problems are extended to a general case with avoiding a consideration of the limiting or boundary cases when the application of this approach starts to be problematic. The partial scattered intensities from aggregate populations are not an orthogonal system, which means that the solutions of the problems related to the series expansion depend on criteria used to judge about correctness and provide stability of the solutions regarding experimental errors. I did not find such kind of criteria in the work. There is no such problem with SEC-SAXS where the partial contributions are directly measured.

So, my recommendation is somewhat between 'decline' and 'major revision'.

→ Again, we appreciate the careful review and comments. We thought that your comments were classified into three points as follows.

1. Generality of AUC-SAS: Choose of a highly-purified sample as a target system.
2. Mathematical back ground: Estimation of scattering intensity of oligomers in higher- q range.
3. Applicable limitation: Estimation of oligomer concentrations.

We would like to reply to each point as follows.

1. Generality of AUC-SAS: Choose of a highly-purified sample as a target system.

As the first trial, our developing AUC-SAS targets a highly-purified sample (less denatured contamination and a small amount of oligomers with low aggregation-degrees) which is usually used for a general SAS measurement. In addition, we are thinking that our next target should be development and/or improvement of this

AUC-SAS method to be applicable for samples not-highly purified. Therefore, we discussed such challenges in a new section “Limitation and improvement of AUC-SAS” in §S7 of SI.

Going back to the issue about the target system, we would like to describe the reason why we concentrated on development of the method targeting a highly-purified sample briefly.

Recently, increasing attention to dynamics and/or function of biomacromolecules in a physiological condition, SAS for biomacromolecules in solution is measured by many biologists, *not only SAS experts but also non-experts*. It should be the primary requirement to obtain a scattering intensity from single biomacromolecules in order to perform a precise structural analysis. Under this situation, in a real SAS research, we usually apply Guinier analysis to an experimental data as the first analytical step. Here, it has been often believed that the sample is a monodispersed solution of monomers/target molecules when the observed scattering intensity shows a linear line in the Guinier plot. However, we have strong concern about this criterion of monodispersity. Figure R3-1 (Fig.S2(b)) is one of representative examples. The sample was highly-purified as usually used in a SAS experiment and the scattering intensity held Guinier approximation: the experimental data showed a linear line in the Guinier plot. On the other hand, AUC clearly indicated that there were several aggregates in the sample (Fig.3). Therefore, even if the sample is highly purified and the scattering profile looks a linear line in the Guinier plot as well, the sample may include a small amount of aggregates and incorrect R_g and $I(0)$ could be given. In other words, only with SAS data, it is very dangerous and difficult to judge whether the sample is monodispersed or not, especially for not SAS expert. Therefore, we added the following sentences in INTRODUCTION,

“In addition, a hidden problem has become appeared^{7,8}. We usually prepare for a purified sample in a structural analysis of a single molecule. However, even though the measured scattering intensity does not show such abnormal upturn and holds Guinier approximation⁷, it could include a small amount of aggregates and they should be removed to obtain correct structure parameters⁸. In other words, it is difficult to judge that there still remains small amount of aggregates in a solution only with SAS: A highly purified sample (Fig.S2(a)) looks nicely holds Guinier approximation, as shown in Fig.S2(b), but it included a small amount of aggregates.”

Again, our main motivations of this present work are to inform this significant problem widely and develop the method to extract a scattering of the target molecule from a whole solution scattering even with aggregates. Therefore, we submit this report to the general journal in biology field, considering the recent broad users of SAS.

Of course, we would like to expand this technique to the more general case such as the sample including relatively more aggregates as described before. For that expansion, we recognize the significance to make mathematical background clear as much as possible and to define the limitation of the present method, which are your suggested concerns #2 and #3. In this revised version, we added the discussion about the concern #2 for a highly-purified sample with the small amount of aggregates and also estimation of the concentration boundary concerned about #3.

Based on the discussion and estimation, as a next target, we would like to extend our method to more general case and then we also would like to submit the result to specialized journal such as Journal of Applied Crystallography as following your advice and assistance.

Fig.R3-1 Guinier plot of scattering intensity for the highly-purified BSA solution selected from Figure 2(b)

2. Mathematical back ground: Estimation of scattering intensity of oligomers in higher- q range.

We discussed the mathematical back ground in the reply to the "Concise remarks (1)" and described in §S4 of SI.

3. Applicable limitation: Estimation of oligomer concentrations.

Following your comments, we discussed the applicable boundary in a new section "Limitation and improvement of AUC-SAS" (SI-§S7). In the "Limitation" part (§S7-1), AUC-SAS was tested for BSA solutions with various amounts of aggregates (BSA2-4). The present protocol offered the correct scattering intensity for the solution containing with 12.4 % of weight fraction of aggregates, while could not for the solution with 18.6 % of weight fraction of aggregates (Fig.S11). Therefore, the present protocol is applicable up to 12 % and the boundary could locate ca. 15%.

In the "Improvement" part (§S7-2), we considered the reason to make the present protocol failed under the higher concentration of aggregates and then presented the possible improvement. The failure of the present protocol under the higher concentration of aggregates could arise from the inaccuracy of $i_{\text{exp}}(0)$ because the experimental scattering intensity does not hold Guinier approximation (please see Fig.S11(d)). The inaccuracy of $i_{\text{exp}}(0)$ causes the wrong $i_1(0)$ which is an important value in Step 5A-1 of the preset AUC-protocol. To obtain more precise $i_1(0)$, we calculated $i_1(0)$ with eq.(S21). This improved approach led the consistent result with SEC-SAXS for the sample even with 18.6 % of weight fraction of aggregates as shown in Fig.S12.

We would like to discuss the further improvement and development of the methodology in our next report, for example, the breakage of the approximation of $i_{1h}(q) \approx i_{\text{exp}}(q)$ described in SI-§S4, at a specialized journal such as Journal of Applied Crystallography.

Concise remarks

1) From the beginning, the authors use an extrapolation procedure for low q -values which is not discussed in the paper. However, the result depends on the smoothness of the scattered intensity in 'high q -value interval'. This smoothness is implicitly assumed for experimental data. Also, it depends on the choice of the minimum cut-off q -value of the 'high q -value interval'. In other words, the procedure is model dependent and, more

important, affected by the choice of the q -value where the ‘high q -value interval’ starts. In Supplementary one can find the calculation of the simplest structure-factor for spherical dimers for one (!) size-value to judge about the aggregation effect on the choice of this interval. No clear criteria based on mathematical consideration are given. It is important that for the general case, such consideration should take into account the restrictions on the finite q -range, as well as possible effects of statistics and background (in case when the absolute scattering is not so high).

→ Thank you for the helpful comments. We made the further discussion of the effect of scattering intensities of oligomers on the approximation $i_{1h}(q_h) \approx i_{mul}(q_h) = i_{exp}(q_h)$ (eq.(3)) in §S4. Here, we describe the summary of the consideration (please see the detail in §S4).

1. We expanded the calculation of scattering intensities of oligomers to trimers and tetramers. However, because it is very difficult to take consideration of all possible oligomers, we adopted the monomer configuration averaged in oligomers as shown in Fig.S3(a). With this configuration, we calculated scattering intensities of two series of oligomers also shown in Fig.S3(b) and (c).

2. We calculated the scattering intensity ratio between monomer and whole system as

$$r(q) \equiv \frac{i_{mul}(q)}{i_1(q)}$$

with the concentrations of demonstrated BSA sample (BSA1). It was

found that $r(q)$ asymptotically approached unity as shown in Fig.S4.

3. We figured out the maximum deviation amplitude $Amp_{r_{max}} = |1 - r(q)|_{q=1.7/R_{g1}} = 1.8\%$ in the higher q range $q > 1.7/R_{g1}$. Therefore, within this error level ca. 1.8% at least, the approximation holds to the minimum cut-off q -value $q \approx 1.7/R_{g1}$. Our initial connection point q_c for the demonstration system was $q_c = 1.9/R_{g1}$, being satisfied with the minimum cut-off value $q \approx 1.7/R_{g1}$ within the maximum error 1.8%. Therefore, we added this mathematical error on the final scattering intensity of monomer.

4. It is meaningful to estimate the dependence of the maximum deviation amplitude $Amp_{r_{max}}$ on the concentration of oligomers. We calculated it as shown in Fig.S13(b). As mentioned in reply #3 (Applicable limitation), the concentration boundary existed around 15%. Therefore, it is supposed that when $Amp_{r_{max}}$ is less than 3%, the present AUC-SAS protocol could be available.

5. As you pointed out, from the viewpoint of the connection of two scattering profiles, we recognized the smooth connection is not mathematically strict. Therefore, we regarded that the smooth connected scattering profile is a just tentative one, and then refined it with the expanded Guinier formula based on Debye function which covers in wider q range (§S5-3). This refinement led to more precise scattering profiles for the demonstrated cases in this study.

2) I am not sure that the proposed procedure for ‘connecting’ Guinier region with ‘high q -value interval’ works well for the molecules with an inhomogeneous inner structure under low contrast. In the general case, there can be a parabolic dependence of $I(q)$ at q approaching zero with a positive coefficient (see, for example, the asymptotic behavior of the basic scattering functions in contrast variation approach in M.V.Avdeev, J. Appl. Cryst. 40, 2007, 56). So, the limits on the homogeneous approximation for single units in the aggregates should be discussed and specified, if possible.

→ Thank you for the comment. As you mentioned, there are cases that Guinier approximation does not hold for the scattering profile with the low contrast in a contrast variation approach. Therefore, we strongly recommended to apply the present AUC-SAS protocol to the solution which holds Guinier approximation enough with high contrast.

In addition, following your comment, we examined AUC-SAS with apoferritin, taking it as an example of a protein with inhomogeneous inner structure. As indicated in the later part of SI-§S6, the present AUC-SAS protocol offered the result consistent with that of SEC-SAXS.

3) I appreciate the comparison with SEC-SAXS. It is a strong evidence for the correctness of the proposed approach. Still, it would be more convincing from the point of view of generalization if the authors make separate simulations of experimental data for model aggregated units of spherical shape with different error levels in typical experimental q -ranges for different unit size scales and treat them in the way proposed. From the comparison with the exact scattering known, one can analyze the systematic deviations of the solutions and their stability with respect to the experimental errors.

→ Thank you for the encouraging comment. Following your suggestion, we compared the AUC-SAS result with the simulated one. Firstly, we simulated scattering intensities of BSA oligomers based on the crystal structure. The dimer structure has been given (PDB code: 4F5S), and the structures of trimer and tetramer were modeled with the dimer as shown in Fig.R3-2(a). Then, we calculated their scattering intensities, $i_{2s}(q)$, $i_{3s}(q)$, and $i_{4s}(q)$ (Fig.R3-2(b)), and we extracted the scattering intensity of monomer $c_1 i_{1s}(q)$ from the experimental one with a following equation.

$$c_1 i_{1s}(q) = c i_{\text{exp}}(q) - c_2 i_{2s}(q) - c_3 i_{3s}(q) - c_4 i_{4s}(q),$$

where $\{c_j\}$ was given with AUC. As shown in Fig.R3-2 (c) and (d), we confirmed $c_1 i_{1s}(q)$ was well consistent with $c_1 i_1(q)$ provided with AUC-SAS.

Fig.R3-2 (a) Monomer, dimer and modeled trimer and tetramer of BSA. The trimer and tetramer are modeled with the crystal structure of dimer (PDB code: 4F5S). (b) Open black circles represent the experimental scattering intensity $c i_{\text{exp}}(q)$. Solid purple, green, and yellow lines show the scattering intensities of dimer ($c_2 i_{2s}(q)$), trimer

$(c_3 i_{3s}(q))$, and tetramer $(c_4 i_{4s}(q))$ with $\{c_j\}$ obtained by AUC, respectively. Comparison of AUC-SAS with the simulation: (c) Scattering intensities of monomer and (d) the Guinier plot. Closed blue and open pink circles are $c_1 i_1(q)$ provided with AUC-SAS and $c_1 i_{1s}(q)$ calculated by monomer, dimer and modeled oligomers, respectively. Solid cyan line represents the scattering intensity calculated with the crystal structure of BSA monomer. Open black circles in the bottom figure of panel (c) are the residuals between $c_1 i_1(q)$ and $c_1 i_{1s}(q)$: $(c_1 i_1(q) - c_1 i_{1s}(q))/c_1 i_1(q)$. Solid blue and pink lines in panel (d) are the least square fitting with the Guinier formula.

Response to Reviewer #4:

This manuscript by Morishima et al. presented a nice method for SAS data analysis by taking advantage of the information of the species in solution from analytical ultracentrifugation (AUC), the AUC-SAS method. Because the experimental conditions of both methods can be the same, the detailed information about the solution composition from AUC, particularly the size and fraction of the aggregates can seemly offer reliable constraints to the data analysis of SAS. This method can be applied to both non-interacting and interacting systems.

For non-interacting system, the key points of this approach are (1) to study the heterogeneous samples with sedimentation velocity analytical ultracentrifugation (SV-AUC); (2) to resolve the size distribution of the sample using the standard $c(s)$ analysis; (3). To decompose the SAS signal by utilizing the size/population information from SV-AUC. The authors also demonstrated that this approach can be applied to study interacting system, by characterization with sedimentation equilibrium in order to extract the fraction of the population for the unbound and bound species.

→ We sincerely appreciate the careful review and understanding of the significance. The following comments have helped us to improve the paper.

Overall, the manuscript is a clear and concise summary of the new AUC-SAS method. Nevertheless, several points should be addressed to improve the clarity of the manuscript. (1). Page4, “Step1..if the scattering intensity deviates from the Guinier approximation..”. What is the threshold/tolerance of the deviation?

→ Thank you for the helpful comment. The criterion of deviation from the Guinier approximation is non-linearity of the experimental data in Guinier plot. Accordingly, the clear deviation of an experimental data from the linear line means that the sample includes a large amount of aggregates. In that case, to force to perform the Guinier fitting, we have to conduct the scattering profile in the very limited linear range, such as very lower q -range. As a result, even though R_g and the molecular weight were obtained, they should be overestimated from their crystal structure data. As an example, we added the deviation of experimental scattering from the Guinier line in Fig.S1 (c). The

increase of deviation in the lower q -range typically means that the solution includes the larger aggregates.

(2). Page5: “Step2: AUC measurement”

The authors should describe why the two particular concentrations, 50 and 100 μM were used.

→ Thank you for the comment. In the previous version, we carried out SE-AUC at two concentrations (50 and 100 μM with the rotation speed of 30000 rpm) to obtain K_D . To improve the reliability of K_D , as you point out in comment (4), the additional SE-AUC measurements were carried out with three concentrations (50, 75, and 100 μM) and three rotation speeds (20000, 30000, and 35000 rpm). We described the detail in §S11. “SE-AUC for determination of K_D and concentration of each component”. The concentrations were chosen to be low enough, where the intermolecular interference effect can be negligible on the scattering profile. As a result, no interference was confirmed because the molecular weights obtained from $i_A(0)$ and $i_B(0)$ for the hHR23b-UBL and PNGase-PUB (Table S4) are consistent with that calculated from their amino acid sequences. Furthermore, the highest volume fraction of protein in the solution subjected to SAXS measurement was 0.002 (200 μM of PNGase-PUB), where the intermolecular interference effect is negligible. (cf. according to the Percus-Yevick model, the significant intermolecular interference arises at higher than ca.0.125 of volume fraction.)

(3). How to distinguish the denatured contaminants and the oligomers from SAS and AUC data?

→ Thank you for the comment. First of all, denatured contaminations should be removed because AUC-SAS is applied to a highly-purified sample as used in a general SAS measurement. It may be difficult to distinguish the denatured oligomer from non-denatured one. However, at least, we are checking the solution about the contaminations with the following procedure.

1. We obtain the s -values for components in a solution with SV-AUC.

2. We calculate their molecular weights M_j for the j -th component in the order of s -values.
3. We check the molecular weights ratio u_j as $u_j \equiv M_j/M_1$ for all components where the ratio is close to an integer j .

The molecular weight is calculated with the average frictional ratio \bar{f}_r over all molecules. Because the frictional ratio f_r of a denatured molecule is supposed to deviate from \bar{f}_r due to the deformation of its shape, the calculated molecular weight also deviates from the correct molecular weight. Therefore, it is expected that the values of u_j ($j \geq 2$) including denatured oligomers deviate from the integers j . We examined this hypothesis as follows.

[Case 1] Solution including small amounts of non-denatured oligomers such as a highly purified sample:

The distributions of s -values of highly purified BSA1 and OVA samples used in this study are shown as black lines in Fig.R4-1 (a) and (b). The calculated values of $\{u_j\}$ are $\{1.00, 2.02, 3.01, 3.97\}$ for the BSA1 solution and $\{1.00, 1.88, 3.04\}$ for the OVA solution, respectively.

[Case 2] Solution including denatured oligomers:

The distributions of s -values of **heated** OVA sample which is expected to be denatured is shown as red line in Fig.R4-1(b). The calculated value of $\{u_j\}$ indicates the deviation from a set of integers, $\{1.00, 1.80, 2.87\}$, as we expected.

We supposed that this procedure, the check of u_j , could work as the first screening of sample if it includes largely denatured contaminations. To confirm the accuracy of this screening, we need to gather more examples and consider about possible assembling shape of oligomers as well.

Fig. R4-1. Sedimentation coefficient distributions for (a) highly-purified BSA and (b) OVA (black: highly-purified and red: heat-denatured) solutions.

(4). There is some limitation of the application to study the weakly bounded complex in the current work.

(i). Sedimentation equilibrium data analysis is an ill-conditioned problem, which is very sensitive to degraded products or any contamination with small particles. Therefore, it is crucial to perform the experiment with multiple concentrations and multiple rotor speeds, so the concentration gradients can be globally analyzed to determine the population of the species with different sizes. In this study, only one rotor speed, 30,000 rpm was used. This limits the accuracy of the data analysis. At least, the authors should perform a reliable error analysis to determine the confidence intervals for K_D of the hetero-association.

→ Thank you for the helpful advice. We carried out the additional SE-AUC measurements, and finally we obtained nine data sets at three concentrations (50, 75, and 100 μM) and three rotation speeds (20000, 30000, and 35000 rpm). Making the global fitting analysis to them, the reliable K_D was provided. The re-estimated result is almost consistent with the previous one. The fitting error was evaluated in the reply to the comment (4-ii).

(ii). Because of the intrinsic limitation of sedimentation equilibrium described above, the authors should consider further analysis for the SAS data using the min and max of the K_D , as well as floating it, to examine whether a better fit can be reached.

→ Thank you for the helpful comment. As you advised, the scattering profiles were calculated for the possible maximum (case 1; $K_D = 255 \mu\text{M}$) and minimum (case 2; $K_D = 199 \mu\text{M}$) of the K_D -values (Fig.R4-2). R_{gAB} , $i_{AB}(0)$, and M_{AB} obtained from the Guinier analysis are consistent between the case 1 and 2 within the errors (Table R4-1). Therefore, the error level of K_D in this study does not affect to the final scattering intensity. For the further improvement, we re-calculated R_{gAB} , $i_{AB}(0)$, M_{AB} and also $i_{AB}(q)$ with considering propagation of the error in K_D (case 3; $K_D = 227 \pm 28 \mu\text{M}$) and the revised values are shown in Table S4, Fig.6(a), and Fig.R4-2.

Fig.R4-2: Purple, cyan, and blue circles represent the $i_{AB}(q)$ provided by using $K_D = 255 \mu\text{M}$ (case 1; possible maximum), $K_D = 199 \mu\text{M}$ (case 2; possible minimum), and $K_D = 227 \pm 28 \mu\text{M}$ (case 3; considering error propagation), respectively. Inset shows their Guinier plots. Solid lines are the least square Guinier fitting lines.

Table R4-1: Gyration radii and forward scatterings obtained from the Guinier analysis.

$K_D / \mu\text{M}$	$R_{gAB} / \text{Å}$	$i_{AB}(0) / 10^{-3} \text{mg}^{-1} \text{cm}^2$	M_{AB} / kDa
255 (case 1)	17.2 ± 0.4	16.0 ± 0.4	21.2 ± 0.6
199 (case 2)	17.0 ± 0.4	15.3 ± 0.4	20.3 ± 0.6
277 ± 28 (case 3)	17.1 ± 0.4	15.7 ± 0.4	20.8 ± 0.6

(5). In the main text, the authors should add a sentence to explain that this is a highly pure interacting system and aggregates are not observed from the individual proteins and their mixture. Therefore, the modeling can be simply based on mass action law.

→ Thank you for the helpful comment. Absence of aggregates was confirmed with SV-AUC (§S10). The aggregation-check with SV-AUC is inevitable in the procedure before SE-AUC. Therefore, as you point out, we added the sentence in page 7 as follows.

"Here, absence of aggregates should be confirmed with SV-AUC (SI-§S10) prior to SE-AUC measurement as a quality check of sample."

Response to Reviewer #2:

In this manuscript, the authors use a new combined experimental technique in the field to obtain precise scattering intensity of biomacromolecules in solution. A combined AUC-SAS analysis is claimed to be a better and more accurate substitute to the current state of the art of utilizing a combination of SEC-SAXS.

I do believe this is a promising technique and will be an important addition in this research field. I do recommend the manuscript for publication after making some changes to make it a better read.

→ We would like to express our sincere appreciation to the reviewer for the encouraging comment. The following comments have helped us to improve the paper.

However, the concept of combining AUC with other techniques is an ongoing research area and I would advise the authors to cite a few recent publications where AUC is combined with other techniques for orthogonal characterization, especially for nanoparticles.

→ Thank you for the helpful advice. Following your comments, we cited the previous studies (ref. 12-14) which applied AUC and SAS to nanoparticles and biomacromolecules. In addition, the previous studies basically used AUC and SAS for the cross check of quality of samples in our knowledge. On the other hand, in our study, AUC and SAS were complementally used to extract the scattering intensity of the target component from that of multi-components system. Therefore, we added the following sentences with above mentioned citations (ref. 12-14) in page 2.

"Therefore, several previous studies used AUC to check the quality of SAS samples¹²⁻¹⁴. On the other hand, we have conceived in the advance to utilize AUC for the "removal": The SAS intensity of multi-component solution can be decomposed into those of components by utilizing the information of their concentration distribution obtained with AUC. "

I had to read quite a few times to understand the algorithm. The section on extracting monomer scattering needs a lot of refinement. It is difficult to shift between the main manuscript and the supplementary information frequently. There is a lot of important information in the SI. Unless there is a restriction of length for the main manuscript, important details like the use of forward scattering, approximations in the high q range etc. should be part of the main manuscript. Please make sure all the variables used are properly explained. The manuscript requires thorough check for typographical errors and good editing.

→ We are very sorry for difficulty to read the manuscript. We revised the explanation about the algorithm and re-arranged its order as follows.

1. Concerning about explanation of intensity in the high q range (new Step 4, please see below), we revised the paragraph of Step 4 to explain the mathematical background more. Even though the further detailed explanation was written in §S4 of SI, we believed that the main logic can be caught by reading the paragraph of Step 4 in the main manuscript.
2. New Figs.4 and 5 were moved from SI to the main manuscript.
3. To make the algorithm clearer, old Step 4 "Scattering profile in low q range" was divided into "Calculation of forward scattering intensity" and "Set of initial scattering intensity".
4. "Calculation of forward scattering intensity" was put on new Step 3, and "Set of initial scattering intensity" was also put on new Step 5A-1.
5. Then old Step 3 "Estimation of scattering intensity in high q range" was moved to new Step 4.
6. Figure 1 was redrawn to follow the above mentioned rearrangements of the algorithm.
7. Old Fig.2 were divided into two figures. New Figs. 2 and 3 were concerned about "Extraction of monomer scattering" and new Fig.6 was about "Extraction of scattering of weakly bounded complex".

This modification on the algorithm makes no effect on the final result.

Two main assumptions are low aggregation degree and the total weight fraction of aggregates to be on the lower side (given approximately as 10 %). The authors need to justify this. The examples given have aggregation degree and relatively weight percentage of aggregates much lower than these values. They need to elaborate more on these concepts and possibly give a range in which the proposed method would work.

→ Thank you for the helpful comments. Following the suggestion, the new section "Limitation and improvement of AUC-SAS" in SI-§S7 was added to discuss the applicable limitation in the aggregation degree (up to 4) and their total weight fractions. In addition, we presented the further possibilities of AUC-SAS to apply to a system with the higher concentration of aggregates.

In the "Limitation" part (§S7-1), AUC-SAS was tested for BSA solutions with various amounts of aggregates (BSA2-4). The present protocol offered the correct scattering intensity for the solution containing with 12.4 % of weight fraction of aggregates, while could not for the solutions with 18.6 % of weight fraction of aggregates (Fig.S11). Therefore, the present protocol is applicable up to 12 % and the boundary could locate ca. 15%.

In the "Improvement" part (§S7-2), we considered the reason to make the present protocol failed under the higher concentration of aggregates and then presented the possible improvement. The failure of the present protocol under the higher concentration of aggregates could arise from the inaccuracy of $i_{\text{exp}}(0)$ because the experimental scattering intensity does not hold Guinier approximation (see Fig.S11(d)). The inaccuracy of $i_{\text{exp}}(0)$ causes the wrong $i_1(0)$ which is an important value in Step 5A-1 of the preset AUC-protocol. As an improved approach to obtain more precise $i_1(0)$, we estimated $i_1(0)$ with eq.(S21). The improved approach led the consistent result with SEC-SAXS for the sample even with 18.6 % of weight fraction of aggregates as shown in Fig.S12. We will make the further discussion including other limitations and improvements in a next paper, for example, the breakage of the approximation of $i_{1h}(q) \approx i_{\text{exp}}(q)$ described in SI-§S4.

Response to Reviewer #3:

The paper 'Integral approach to biomacromolecular structure by analytical-ultracentrifugation and small-angle scattering' by K. Morishima et al. presents a semi-empirical approach to correct SAS curves from biological molecules in solutions regarding the aggregation effect. The correction is made basing on an additional measurement of partial fractions of aggregates by analytical ultra-centrifuging. This way is considered to some extent as an alternative to SEC-SAXS. It is demonstrated that the approach works well for several protein solutions, which is proved by consistency with the data of SEC-SAXS. The motivation is claimed well. The problem of the protein aggregation in SAS experiments is old. The problem is actual and not only from the point of view of parasitic effects. See, for example, a very recent publication on the study of oligomeric solutions of proteins at the pre-crystallization stage by SAXS [Marchenkova MA, et al., J Biomol Struct Dyn. 2019. doi: 10.1080/07391102.2019.1649195].

→ First of all, we sincerely appreciate your careful review and comments. All comments are helpful for revise of the manuscript.

After carefully reading the suggested paper, we recognize that there are meaningful aggregates in a multi-component solution in these days. In this study, we did not examine an AUC-SAS protocol with an oligomeric solution of proteins at the pre-crystallization stage in the suggested paper directly, but demonstrated the protocol for the similar system (nucleosome in SI-§S13). Therefore, we added following sentences with the reference you suggested in page 8.

"In these days, the structural analysis of oligomers¹⁷ in a multi-component solution attracts attention as well as that of monomer. For such cases, AUC-SAS is applicable if the all scattering intensities except for the concerned component are individually available: the demonstration of AUC-SAS for nucleosome could be one of examples (SI-§S13)."

General remark

I am afraid the authors have not fully convinced me that the approach proposed is general and works well in all cases, not only for the solutions considered in the paper.

The approach lacks substantial mathematical background which should first be presented and discussed in details, may be, in a more specialized journal such as Journal of Applied Crystallography before it is considered to be a generalized method (such as, for example, indirect Fourier transform). I would definitely support the approach if it were used as a complementary part of some complex characterization of protein solutions close to those of BSA. Here, I have an impression that numerical solutions of partial problems are extended to a general case with avoiding a consideration of the limiting or boundary cases when the application of this approach starts to be problematic. The partial scattered intensities from aggregate populations are not an orthogonal system, which means that the solutions of the problems related to the series expansion depend on criteria used to judge about correctness and provide stability of the solutions regarding experimental errors. I did not find such kind of criteria in the work. There is no such problem with SEC-SAXS where the partial contributions are directly measured.

So, my recommendation is somewhat between 'decline' and 'major revision'.

→ Again, we appreciate the careful review and comments. We thought that your comments were classified into three points as follows.

1. Generality of AUC-SAS: Choose of a highly-purified sample as a target system.
2. Mathematical back ground: Estimation of scattering intensity of oligomers in higher- q range.
3. Applicable limitation: Estimation of oligomer concentrations.

We would like to reply to each point as follows.

1. Generality of AUC-SAS: Choose of a highly-purified sample as a target system.

As the first trial, our developing AUC-SAS targets a highly-purified sample (less denatured contamination and a small amount of oligomers with low aggregation-degrees) which is usually used for a general SAS measurement. In addition, we are thinking that our next target should be development and/or improvement of this

AUC-SAS method to be applicable for samples not-highly purified. Therefore, we discussed such challenges in a new section “Limitation and improvement of AUC-SAS” in §S7 of SI.

Going back to the issue about the target system, we would like to describe the reason why we concentrated on development of the method targeting a highly-purified sample briefly.

Recently, increasing attention to dynamics and/or function of biomacromolecules in a physiological condition, SAS for biomacromolecules in solution is measured by many biologists, *not only SAS experts but also non-experts*. It should be the primary requirement to obtain a scattering intensity from single biomacromolecules in order to perform a precise structural analysis. Under this situation, in a real SAS research, we usually apply Guinier analysis to an experimental data as the first analytical step. Here, it has been often believed that the sample is a monodispersed solution of monomers/target molecules when the observed scattering intensity shows a linear line in the Guinier plot. However, we have strong concern about this criterion of monodispersity. Figure R3-1 (Fig.S2(b)) is one of representative examples. The sample was highly-purified as usually used in a SAS experiment and the scattering intensity held Guinier approximation: the experimental data showed a linear line in the Guinier plot. On the other hand, AUC clearly indicated that there were several aggregates in the sample (Fig.3). Therefore, even if the sample is highly purified and the scattering profile looks a linear line in the Guinier plot as well, the sample may include a small amount of aggregates and incorrect R_g and $I(0)$ could be given. In other words, only with SAS data, it is very dangerous and difficult to judge whether the sample is monodispersed or not, especially for not SAS expert. Therefore, we added the following sentences in INTRODUCTION,

“In addition, a hidden problem has become appeared^{7,8}. We usually prepare for a purified sample in a structural analysis of a single molecule. However, even though the measured scattering intensity does not show such abnormal upturn and holds Guinier approximation⁷, it could include a small amount of aggregates and they should be removed to obtain correct structure parameters⁸. In other words, it is difficult to judge that there still remains small amount of aggregates in a solution only with SAS: A highly purified sample (Fig.S2(a)) looks nicely holds Guinier approximation, as shown in Fig.S2(b), but it included a small amount of aggregates.”

Again, our main motivations of this present work are to inform this significant problem widely and develop the method to extract a scattering of the target molecule from a whole solution scattering even with aggregates. Therefore, we submit this report to the general journal in biology field, considering the recent broad users of SAS.

Of course, we would like to expand this technique to the more general case such as the sample including relatively more aggregates as described before. For that expansion, we recognize the significance to make mathematical background clear as much as possible and to define the limitation of the present method, which are your suggested concerns #2 and #3. In this revised version, we added the discussion about the concern #2 for a highly-purified sample with the small amount of aggregates and also estimation of the concentration boundary concerned about #3.

Based on the discussion and estimation, as a next target, we would like to extend our method to more general case and then we also would like to submit the result to specialized journal such as Journal of Applied Crystallography as following your advice and assistance.

Fig.R3-1 Guinier plot of scattering intensity for the highly-purified BSA solution selected from Figure 2(b)

2. Mathematical back ground: Estimation of scattering intensity of oligomers in higher- q range.

We discussed the mathematical back ground in the reply to the "Concise remarks (1)" and described in §S4 of SI.

3. Applicable limitation: Estimation of oligomer concentrations.

Following your comments, we discussed the applicable boundary in a new section "Limitation and improvement of AUC-SAS" (SI-§S7). In the "Limitation" part (§S7-1), AUC-SAS was tested for BSA solutions with various amounts of aggregates (BSA2-4). The present protocol offered the correct scattering intensity for the solution containing with 12.4 % of weight fraction of aggregates, while could not for the solution with 18.6 % of weight fraction of aggregates (Fig.S11). Therefore, the present protocol is applicable up to 12 % and the boundary could locate ca. 15%.

In the "Improvement" part (§S7-2), we considered the reason to make the present protocol failed under the higher concentration of aggregates and then presented the possible improvement. The failure of the present protocol under the higher concentration of aggregates could arise from the inaccuracy of $i_{\text{exp}}(0)$ because the experimental scattering intensity does not hold Guinier approximation (please see Fig.S11(d)). The inaccuracy of $i_{\text{exp}}(0)$ causes the wrong $i_1(0)$ which is an important value in Step 5A-1 of the preset AUC-protocol. To obtain more precise $i_1(0)$, we calculated $i_1(0)$ with eq.(S21). This improved approach led the consistent result with SEC-SAXS for the sample even with 18.6 % of weight fraction of aggregates as shown in Fig.S12.

We would like to discuss the further improvement and development of the methodology in our next report, for example, the breakage of the approximation of $i_{1h}(q) \approx i_{\text{exp}}(q)$ described in SI-§S4, at a specialized journal such as Journal of Applied Crystallography.

Concise remarks

1) From the beginning, the authors use an extrapolation procedure for low q-values which is not discussed in the paper. However, the result depends on the smoothness of the scattered intensity in 'high q-value interval'. This smoothness is implicitly assumed for experimental data. Also, it depends on the choice of the minimum cut-off q-value of the 'high q-value interval'. In other words, the procedure is model dependent and, more

important, affected by the choice of the q -value where the ‘high q -value interval’ starts. In Supplementary one can find the calculation of the simplest structure-factor for spherical dimers for one (!) size-value to judge about the aggregation effect on the choice of this interval. No clear criteria based on mathematical consideration are given. It is important that for the general case, such consideration should take into account the restrictions on the finite q -range, as well as possible effects of statistics and background (in case when the absolute scattering is not so high).

→ Thank you for the helpful comments. We made the further discussion of the effect of scattering intensities of oligomers on the approximation $i_{1h}(q_h) \approx i_{mul}(q_h) = i_{exp}(q_h)$ (eq.(3)) in §S4. Here, we describe the summary of the consideration (please see the detail in §S4).

1. We expanded the calculation of scattering intensities of oligomers to trimers and tetramers. However, because it is very difficult to take consideration of all possible oligomers, we adopted the monomer configuration averaged in oligomers as shown in Fig.S3(a). With this configuration, we calculated scattering intensities of two series of oligomers also shown in Fig.S3(b) and (c).

2. We calculated the scattering intensity ratio between monomer and whole system as

$$r(q) \equiv \frac{i_{mul}(q)}{i_1(q)}$$

with the concentrations of demonstrated BSA sample (BSA1). It was

found that $r(q)$ asymptotically approached unity as shown in Fig.S4.

3. We figured out the maximum deviation amplitude $Amp_{r_{max}} = |1 - r(q)|_{q=1.7/R_{g1}} = 1.8\%$ in the higher q range $q > 1.7/R_{g1}$. Therefore, within this error level ca. 1.8% at least, the approximation holds to the minimum cut-off q -value $q \approx 1.7/R_{g1}$. Our initial connection point q_c for the demonstration system was $q_c = 1.9/R_{g1}$, being satisfied with the minimum cut-off value $q \approx 1.7/R_{g1}$ within the maximum error 1.8%. Therefore, we added this mathematical error on the final scattering intensity of monomer.

4. It is meaningful to estimate the dependence of the maximum deviation amplitude $Amp_{r_{max}}$ on the concentration of oligomers. We calculated it as shown in Fig.S13(b). As mentioned in reply #3 (Applicable limitation), the concentration boundary existed around 15%. Therefore, it is supposed that when $Amp_{r_{max}}$ is less than 3%, the present AUC-SAS protocol could be available.

5. As you pointed out, from the viewpoint of the connection of two scattering profiles, we recognized the smooth connection is not mathematically strict. Therefore, we regarded that the smooth connected scattering profile is a just tentative one, and then refined it with the expanded Guinier formula based on Debye function which covers in wider q range (§S5-3). This refinement led to more precise scattering profiles for the demonstrated cases in this study.

2) I am not sure that the proposed procedure for ‘connecting’ Guinier region with ‘high q -value interval’ works well for the molecules with an inhomogeneous inner structure under low contrast. In the general case, there can be a parabolic dependence of $I(q)$ at q approaching zero with a positive coefficient (see, for example, the asymptotic behavior of the basic scattering functions in contrast variation approach in M.V.Avdeev, J. Appl. Cryst. 40, 2007, 56). So, the limits on the homogeneous approximation for single units in the aggregates should be discussed and specified, if possible.

→ Thank you for the comment. As you mentioned, there are cases that Guinier approximation does not hold for the scattering profile with the low contrast in a contrast variation approach. Therefore, we strongly recommended to apply the present AUC-SAS protocol to the solution which holds Guinier approximation enough with high contrast.

In addition, following your comment, we examined AUC-SAS with apoferritin, taking it as an example of a protein with inhomogeneous inner structure. As indicated in the later part of SI-§S6, the present AUC-SAS protocol offered the result consistent with that of SEC-SAXS.

3) I appreciate the comparison with SEC-SAXS. It is a strong evidence for the correctness of the proposed approach. Still, it would be more convincing from the point of view of generalization if the authors make separate simulations of experimental data for model aggregated units of spherical shape with different error levels in typical experimental q -ranges for different unit size scales and treat them in the way proposed. From the comparison with the exact scattering known, one can analyze the systematic deviations of the solutions and their stability with respect to the experimental errors.

→ Thank you for the encouraging comment. Following your suggestion, we compared the AUC-SAS result with the simulated one. Firstly, we simulated scattering intensities of BSA oligomers based on the crystal structure. The dimer structure has been given (PDB code: 4F5S), and the structures of trimer and tetramer were modeled with the dimer as shown in Fig.R3-2(a). Then, we calculated their scattering intensities, $i_{2s}(q)$, $i_{3s}(q)$, and $i_{4s}(q)$ (Fig.R3-2(b)), and we extracted the scattering intensity of monomer $c_1 i_{1s}(q)$ from the experimental one with a following equation.

$$c_1 i_{1s}(q) = c i_{\text{exp}}(q) - c_2 i_{2s}(q) - c_3 i_{3s}(q) - c_4 i_{4s}(q),$$

where $\{c_j\}$ was given with AUC. As shown in Fig.R3-2 (c) and (d), we confirmed $c_1 i_{1s}(q)$ was well consistent with $c_1 i_1(q)$ provided with AUC-SAS.

Fig.R3-2 (a) Monomer, dimer and modeled trimer and tetramer of BSA. The trimer and tetramer are modeled with the crystal structure of dimer (PDB code: 4F5S). (b) Open black circles represent the experimental scattering intensity $c i_{\text{exp}}(q)$. Solid purple, green, and yellow lines show the scattering intensities of dimer ($c_2 i_{2s}(q)$), trimer

$(c_3 i_{3s}(q))$, and tetramer $(c_4 i_{4s}(q))$ with $\{c_j\}$ obtained by AUC, respectively. Comparison of AUC-SAS with the simulation: (c) Scattering intensities of monomer and (d) the Guinier plot. Closed blue and open pink circles are $c_1 i_1(q)$ provided with AUC-SAS and $c_1 i_{1s}(q)$ calculated by monomer, dimer and modeled oligomers, respectively. Solid cyan line represents the scattering intensity calculated with the crystal structure of BSA monomer. Open black circles in the bottom figure of panel (c) are the residuals between $c_1 i_1(q)$ and $c_1 i_{1s}(q)$: $(c_1 i_1(q) - c_1 i_{1s}(q))/c_1 i_1(q)$. Solid blue and pink lines in panel (d) are the least square fitting with the Guinier formula.

Response to Reviewer #4:

This manuscript by Morishima et al. presented a nice method for SAS data analysis by taking advantage of the information of the species in solution from analytical ultracentrifugation (AUC), the AUC-SAS method. Because the experimental conditions of both methods can be the same, the detailed information about the solution composition from AUC, particularly the size and fraction of the aggregates can seemly offer reliable constraints to the data analysis of SAS. This method can be applied to both non-interacting and interacting systems.

For non-interacting system, the key points of this approach are (1) to study the heterogeneous samples with sedimentation velocity analytical ultracentrifugation (SV-AUC); (2) to resolve the size distribution of the sample using the standard $c(s)$ analysis; (3). To decompose the SAS signal by utilizing the size/population information from SV-AUC. The authors also demonstrated that this approach can be applied to study interacting system, by characterization with sedimentation equilibrium in order to extract the fraction of the population for the unbound and bound species.

→ We sincerely appreciate the careful review and understanding of the significance. The following comments have helped us to improve the paper.

Overall, the manuscript is a clear and concise summary of the new AUC-SAS method. Nevertheless, several points should be addressed to improve the clarity of the manuscript. (1). Page4, “Step1..if the scattering intensity deviates from the Guinier approximation..”. What is the threshold/tolerance of the deviation?

→ Thank you for the helpful comment. The criterion of deviation from the Guinier approximation is non-linearity of the experimental data in Guinier plot. Accordingly, the clear deviation of an experimental data from the linear line means that the sample includes a large amount of aggregates. In that case, to force to perform the Guinier fitting, we have to conduct the scattering profile in the very limited linear range, such as very lower q -range. As a result, even though R_g and the molecular weight were obtained, they should be overestimated from their crystal structure data. As an example, we added the deviation of experimental scattering from the Guinier line in Fig.S1 (c). The

increase of deviation in the lower q -range typically means that the solution includes the larger aggregates.

(2). Page5: “Step2: AUC measurement”

The authors should describe why the two particular concentrations, 50 and 100 μM were used.

→ Thank you for the comment. In the previous version, we carried out SE-AUC at two concentrations (50 and 100 μM with the rotation speed of 30000 rpm) to obtain K_D . To improve the reliability of K_D , as you point out in comment (4), the additional SE-AUC measurements were carried out with three concentrations (50, 75, and 100 μM) and three rotation speeds (20000, 30000, and 35000 rpm). We described the detail in §S11. “SE-AUC for determination of K_D and concentration of each component”. The concentrations were chosen to be low enough, where the intermolecular interference effect can be negligible on the scattering profile. As a result, no interference was confirmed because the molecular weights obtained from $i_A(0)$ and $i_B(0)$ for the hHR23b-UBL and PNGase-PUB (Table S4) are consistent with that calculated from their amino acid sequences. Furthermore, the highest volume fraction of protein in the solution subjected to SAXS measurement was 0.002 (200 μM of PNGase-PUB), where the intermolecular interference effect is negligible. (cf. according to the Percus-Yevick model, the significant intermolecular interference arises at higher than ca.0.125 of volume fraction.)

(3). How to distinguish the denatured contaminants and the oligomers from SAS and AUC data?

→ Thank you for the comment. First of all, denatured contaminations should be removed because AUC-SAS is applied to a highly-purified sample as used in a general SAS measurement. It may be difficult to distinguish the denatured oligomer from non-denatured one. However, at least, we are checking the solution about the contaminations with the following procedure.

1. We obtain the s -values for components in a solution with SV-AUC.

2. We calculate their molecular weights M_j for the j -th component in the order of s -values.
3. We check the molecular weights ratio u_j as $u_j \equiv M_j/M_1$ for all components where the ratio is close to an integer j .

The molecular weight is calculated with the average frictional ratio \bar{f}_r over all molecules. Because the frictional ratio f_r of a denatured molecule is supposed to deviate from \bar{f}_r due to the deformation of its shape, the calculated molecular weight also deviates from the correct molecular weight. Therefore, it is expected that the values of u_j ($j \geq 2$) including denatured oligomers deviate from the integers j . We examined this hypothesis as follows.

[Case 1] Solution including small amounts of non-denatured oligomers such as a highly purified sample:

The distributions of s -values of highly purified BSA1 and OVA samples used in this study are shown as black lines in Fig.R4-1 (a) and (b). The calculated values of $\{u_j\}$ are $\{1.00, 2.02, 3.01, 3.97\}$ for the BSA1 solution and $\{1.00, 1.88, 3.04\}$ for the OVA solution, respectively.

[Case 2] Solution including denatured oligomers:

The distributions of s -values of **heated** OVA sample which is expected to be denatured is shown as red line in Fig.R4-1(b). The calculated value of $\{u_j\}$ indicates the deviation from a set of integers, $\{1.00, 1.80, 2.87\}$, as we expected.

We supposed that this procedure, the check of u_j , could work as the first screening of sample if it includes largely denatured contaminations. To confirm the accuracy of this screening, we need to gather more examples and consider about possible assembling shape of oligomers as well.

Fig. R4-1. Sedimentation coefficient distributions for (a) highly-purified BSA and (b) OVA (black: highly-purified and red: heat-denatured) solutions.

(4). There is some limitation of the application to study the weakly bounded complex in the current work.

(i). Sedimentation equilibrium data analysis is an ill-conditioned problem, which is very sensitive to degraded products or any contamination with small particles. Therefore, it is crucial to perform the experiment with multiple concentrations and multiple rotor speeds, so the concentration gradients can be globally analyzed to determine the population of the species with different sizes. In this study, only one rotor speed, 30,000 rpm was used. This limits the accuracy of the data analysis. At least, the authors should perform a reliable error analysis to determine the confidence intervals for K_D of the hetero-association.

→ Thank you for the helpful advice. We carried out the additional SE-AUC measurements, and finally we obtained nine data sets at three concentrations (50, 75, and 100 μM) and three rotation speeds (20000, 30000, and 35000 rpm). Making the global fitting analysis to them, the reliable K_D was provided. The re-estimated result is almost consistent with the previous one. The fitting error was evaluated in the reply to the comment (4-ii).

(ii). Because of the intrinsic limitation of sedimentation equilibrium described above, the authors should consider further analysis for the SAS data using the min and max of the K_D , as well as floating it, to examine whether a better fit can be reached.

→ Thank you for the helpful comment. As you advised, the scattering profiles were calculated for the possible maximum (case 1; $K_D = 255 \mu\text{M}$) and minimum (case 2; $K_D = 199 \mu\text{M}$) of the K_D -values (Fig.R4-2). R_{gAB} , $i_{AB}(0)$, and M_{AB} obtained from the Guinier analysis are consistent between the case 1 and 2 within the errors (Table R4-1). Therefore, the error level of K_D in this study does not affect to the final scattering intensity. For the further improvement, we re-calculated R_{gAB} , $i_{AB}(0)$, M_{AB} and also $i_{AB}(q)$ with considering propagation of the error in K_D (case 3; $K_D = 227 \pm 28 \mu\text{M}$) and the revised values are shown in Table S4, Fig.6(a), and Fig.R4-2.

Fig.R4-2: Purple, cyan, and blue circles represent the $i_{AB}(q)$ provided by using $K_D = 255 \mu\text{M}$ (case 1; possible maximum), $K_D = 199 \mu\text{M}$ (case 2; possible minimum), and $K_D = 227 \pm 28 \mu\text{M}$ (case 3; considering error propagation), respectively. Inset shows their Guinier plots. Solid lines are the least square Guinier fitting lines.

Table R4-1: Gyration radii and forward scatterings obtained from the Guinier analysis.

$K_D / \mu\text{M}$	$R_{gAB} / \text{Å}$	$i_{AB}(0) / 10^{-3} \text{mg}^{-1} \text{cm}^2$	M_{AB} / kDa
255 (case 1)	17.2 ± 0.4	16.0 ± 0.4	21.2 ± 0.6
199 (case 2)	17.0 ± 0.4	15.3 ± 0.4	20.3 ± 0.6
277 ± 28 (case 3)	17.1 ± 0.4	15.7 ± 0.4	20.8 ± 0.6

(5). In the main text, the authors should add a sentence to explain that this is a highly pure interacting system and aggregates are not observed from the individual proteins and their mixture. Therefore, the modeling can be simply based on mass action law.

→ Thank you for the helpful comment. Absence of aggregates was confirmed with SV-AUC (§S10). The aggregation-check with SV-AUC is inevitable in the procedure before SE-AUC. Therefore, as you point out, we added the sentence in page 7 as follows.

"Here, absence of aggregates should be confirmed with SV-AUC (SI-§S10) prior to SE-AUC measurement as a quality check of sample."

REVIEWERS' COMMENTS:

Reviewer #1 (Remarks to the Author):

The authors have largely addressed my concerns. I still doubt the general applicability of the approach to fast interactions or to the usual approach to collecting SAXS data at synchrotron facilities. I am still unsure as to why this is a preferred approach to in-line SEC. However, I agree that the approach is relatively sound and is publishable in its current state.

Reviewer #3 (Remarks to the Author):

I am rather impressed with the detailed answers to all reviewers' comments. The work has been significantly improved, thus making reading easier. Definitely, it has a 'zest' that the authors want to stake out a claim for. It is an idea to use the data of AUC to extract, first, the forward scattering contribution of monomers from the total $i(0)$, which can be well done from the mathematical point of view. At the next step, the monomer contribution over 'low-q range' is considered in terms of the extended Guiner approximation with the search of the q-point connecting this range with the 'high-q range', a range which is common for all aggregate populations and is purely determined by the scattering from monomers. As a result, this procedure makes it possible to eliminate the aggregation effect without modelling scattering by monomers, i.e. the same result, which SEC gives after physical separation of the mixture.

The main concern of the reviewer was about the universality of this semi-empirical approach for the case of an arbitrary distribution of the aggregate populations, as well as for the case of non-homogenous monomers with a non-standard behavior in the Guinier region. In response, the authors have restricted the conditions for their approach making an accent to practical application for typical protein solutions which show comparatively small aggregation with a continuously decreasing distribution function for the aggregation number. It looks reasonable, while not free of the risks of the wrong work for 'non-standard' aggregate distributions. To some extent, the authors plan to study the potential developing of the algorithm proposed in a separate paper covering more technical aspects and general mathematical background. As a whole, I am satisfied with the responses. The paper is publishable.

Reviewer #4 (Remarks to the Author):

The authors have addressed the major points raised by the reviewers in a point-by-point manner in the rebuttal. This revised manuscript has been significantly improved from the original one. With the addition of new data, analysis and detailed information of the procedures, particularly the criterion, threshold and conditions for experiments and data interpretation, the authors were able to present this new method clearly. I think the science community will be benefited by this study and suggest it for publication.

Response to Reviewer #1:

Thank you for all your comments and suggestions. They make our paper incredibly improved. About your concern, “I still doubt the general applicability of the approach to fast interactions or to the usual approach to collecting SAXS data at synchrotron facilities. I am still unsure as to why this is a preferred approach to in-line SEC.”

We think that AUC-SAS is a complementary method for SR-SEC-SAS, especially laboratory-based SAXS and SANS. Then, we added the following sentence in the end of “DISCUSSION”:

“Finally, we would like to remark the followings. AUC-SAS does not require the very high intensity beam for a sample-flow experiment such as SEC-SAXS. Therefore, AUC-SAS has a potential to be a complementary method for a laboratory-based SAXS and a standard SANS to synchrotron-based SEC-SAXS for structural analysis of biomacromolecules in solution. ”

Response to Reviewer #3:

Thank you for all your comments and suggestions. They make our paper incredibly improved. Now, we are making a software for AUC-SAS. Then, after finishing the software, we will submit the paper which focused on the technical part

Response to Reviewer #4:

Thank you for all your comments and suggestions. Especially by your comments concerning about the criterion, threshold and conditions for experiments and data interpretation, we made this paper incredibly improved.